# Phagocytosis of *Plasmodium falciparum* ring-stage parasites predicts protection against malaria

Fauzia K. Musasia[1], Irene N. Nkumama[1,2], Roland Frank[1], Victor Kipkemboi[1,3], Martin Schneider[4], Kennedy Mwai [2,5], Dennis O. Odera[1,2], Micha Rosenkranz[1], Kristin Fürle[1], Domitila Kimani[2], James Tuju[2], Patricia Njuguna[2], Mainga Hamaluba [2], Melissa C. Kapulu[2], Hedda Wardemann [6], CHMI-SIKA Study Team* & Faith H. A. Osier [1,2,7✉]

Ring-infected erythrocytes are the predominant asexual stage in the peripheral circulation but are rarely investigated in the context of acquired immunity against *Plasmodium falciparum* malaria. Here we compare antibody-dependent phagocytosis of ring-infected parasite cultures in samples from a controlled human malaria infection (CHMI) study (NCT02739763). Protected volunteers did not develop clinical symptoms, maintained parasitaemia below a predefined threshold of 500 parasites/μl and were not treated until the end of the study. Antibody-dependent phagocytosis of both ring-infected and uninfected erythrocytes from parasite cultures was strongly correlated with protection. A surface proteomic analysis revealed the presence of merozoite proteins including erythrocyte binding antigen-175 and −140 on ring-infected and uninfected erythrocytes, providing an additional antibody-mediated protective mechanism for their activity beyond invasion-inhibition. Competition phagocytosis assays support the hypothesis that merozoite antigens are the key mediators of this functional activity. Targeting ring-stage parasites may contribute to the control of parasitaemia and prevention of clinical malaria.

[1] Centre for Infectious Diseases, Heidelberg University Hospital, Heidelberg, Germany. [2] Centre for Geographic Medicine Research (Coast), Kenya Medical Research Institute (KEMRI)-Wellcome Trust Research Programme, Kilifi, Kenya. [3] Department of Biotechnology, Hochschule Rhein-Waal, Kleve, Germany. [4] Genomics and Proteomics Core Facility, German Cancer Research Center, Heidelberg, Germany. [5] Epidemiology and Biostatistics Division, School of Public Health, University of the Witwatersrand, Johannesburg, South Africa. [6] Division of B Cell Immunology, German Cancer Research Center, Heidelberg, Germany. [7] Department of Life Sciences, Imperial College London, London, UK. *A list of authors and their affiliations appears at the end of the paper. ✉email: f.osier@imperial.ac.uk

Plasmodium species have a complex life cycle that alternates between the female *Anopheles* mosquito and the vertebrate host[1]. The blood stage of the *P. falciparum* life cycle is responsible for the clinical symptoms of malaria and ring-infected erythrocytes (rIEs) are the predominant parasite forms detected in the peripheral circulation[2]. They are thought to contribute to sequestration in severe malaria either directly through cytoadherence to endothelial receptors[3,4], or indirectly via retention in organs due to their reduced deformability[5,6]. Interestingly, the ring-surface proteins that bind to endothelial receptors such as RSP2 have also been detected on uninfected erythrocytes (uEs)[3]. This labelling of uEs leaves them vulnerable to antibody detection and destruction and was postulated to contribute to malarial anaemia[7]. However, antibodies against the same proteins were shown to mediate protection in immunised monkeys and to inhibit parasite growth in vitro[8].

In the context of naturally acquired immunity to malaria, rIEs are understudied. Malaria-immune adults often harbour asymptomatic *P. falciparum* infections in which rIEs are typically detected in blood smears[9]. Parasitaemia is usually low, suggesting that active immune mechanisms limit the exponential parasite multiplication that characterises symptomatic malaria. Ring-surface proteins are recognised by malaria-exposed sera indicating that they are not immunologically inert and may be targets of protective immunity[10,11]. While antibodies to other parasite stages including merozoites and mature-infected erythrocytes (mIEs) have been associated with protection from clinical episodes of malaria[12], similar data on rIEs are lacking. Antibodies targeting rIEs were shown to mediate opsonic phagocytosis[10,11,13,14] but the impact of this activity has neither been analysed in gold standard prospective cohort studies, nor investigated in contemporary human malaria challenge studies.

We utilised a unique experimental study in which we analysed immune responses against rIEs in volunteers who developed either asymptomatic or symptomatic infections following a malaria challenge[15]. We asked whether antibody-dependent phagocytosis of rIEs was correlated with protection, defined as the ability to remain asymptomatic following a standard challenge intravenous inoculum with *P. falciparum* sporozoites. We demonstrate that antibody-dependent phagocytosis of rIEs and uninfected erythrocytes (uEs) from ring cultures was strongly correlated with protection. Contrary to the suggestion that this may lead to anaemia, volunteers with high levels of phagocytosis had significantly higher haemoglobin levels at the end of the study compared to those with low levels. A surface proteomic analysis revealed the presence of merozoite proteins including erythrocyte binding antigen-175 and −140 on ring-infected and uninfected erythrocytes, providing an additional antibody-mediated protective mechanism for their activity beyond invasion-inhibition.

## Results
**Detection of antibody binding to ring-infected erythrocytes (rIEs).** We tested whether malaria-immune plasma bound to the surface of rIEs using immunofluorescence assays (IFA) and flow cytometry. Parasite invasion of erythrocytes is mediated through multiple receptors including Glycophorin A[16]. We therefore used a Fluorescein Isothiocyanate (FITC) anti-human CD235a (Glycophorin A) antibody (Biolegend UK Limited) in a 1:500 dilution as a positive control to localise antibody binding to the surface of rIEs in IFAs. Alexa Fluor® 647 anti-human IgG Fc antibody (Biolegend UK Limited) at a 1:500 dilution was used for secondary staining in IFAs with malaria-immune plasma at a 1:1000 dilution. Mature stage parasites express variant surface antigens on the surface of infected parasites that are known to be antibody targets of protective immunity against malaria[17]. We compared

binding to rIEs to that observed against erythrocytes infected with mature stage parasites (mIEs). We also assessed binding to rIEs in malaria-naive plasma (1:1000 dilution) and using only the secondary antibodies above (without immune plasma) as negative controls. In the presence of malaria-immune plasma, we observed modest to low levels of antibody binding to rIEs compared to mIEs (Fig. 1). In contrast, there was negligible detection of antibody binding to rIEs following opsonization with malaria-naive plasma or using only secondary antibody as a negative control. We observed similar results in flow cytometry experiments using Allophycocyanin (APC) anti-human CD235a (Glycophorin A) secondary antibodies (Biolegend UK Limited) diluted to 1:50 and human plasma samples at 1:12.5 dilution). The proportion of rIEs binding to malaria-immune plasma increased from 0.034 and 0.025 for malaria-naive and secondary antibody controls to 2.31% and 35.4% for rIEs and mIEs, respectively (Supplementary Fig. 1).

The merozoite protein EBA-175 has been localised to the surface of rIEs[18]. It has been previously reported that monoclonal antibodies (mAbs) against EBA-175 recognized segmenters and merozoites in mature schizonts but not in early rings[19]. We tested mAbs R217 and R218 and similarly could not detect binding to the surface of rIEs (Supplementary Fig. 2).

**Antibody-binding to rIEs is dose-dependent, correlated between field and laboratory isolates and with responses to other parasite stages.** Given the relatively modest level of antibody detection on the surface of rIEs, we asked whether this signal would nevertheless suffice to discriminate between positive and negative responders in human samples and provide a range of responses for robust statistical analyses. We used samples from Kenyan adults in the Junju sublocation of Kilifi County (see methods). First, we show that antibody binding to the surface of rIEs is dose-dependent (Fig. 2A). Next, to support the potential relevance of antibody-binding to rIEs using a long-term adapted laboratory isolate (FCR-3[3], we tested whether this was correlated to similar data using a recently adapted *P. falciparum* field isolate from Kilifi, Kenya (Isolate P0000072, Biobank, KEMRI-Wellcome Trust Research Program). Antibody-binding to rIEs using the FCR-3 strain the and the field isolate were highly correlated $R = 0.8$, $P < 0.0001$ (Fig. 2B). We then compared antibody against rIEs by flow cytometry with that to mIEs and whole merozoite extract prepared from the NF54 strain used in the CHMI study. Antibody binding to rIEs was highly correlated with that against mIEs ($R = 0.8$, $P < 0.0001$) and merozoites ($R = 0.8$, $P < 0.0001$), Fig. 2C, D. These data suggested that antibody-binding to rIEs may be useful for discriminating clinical outcomes in the CHMI study.

**Phagocytosis of ring-infected erythrocytes predicts protection against malaria.** We asked whether phagocytosis of ring-infected erythrocytes was correlated with protection in the controlled human malaria infection (CHMI) study. The endpoints were clinical symptoms of malaria with any evidence of malaria parasites by blood film positivity or parasitaemia >500 parasites/μl, both of which warranted immediate treatment[15]. We quantified the level of phagocytosis in samples collected on the day before challenge (C-1) and refer to this as the relative phagocytosis index (the percentage of THP-1 human monocytes that had ingested merozoites[20]. Volunteers who were not treated had a significantly higher phagocytosis index of rIEs compared to those requiring treatment ($P = 0.0003$; Fig. 3A). Treated volunteers were further classified into those who developed fever and those who did not (febrile versus non-febrile). Untreated volunteers were subclassified based on parasite detection by PCR into PCR-ve and

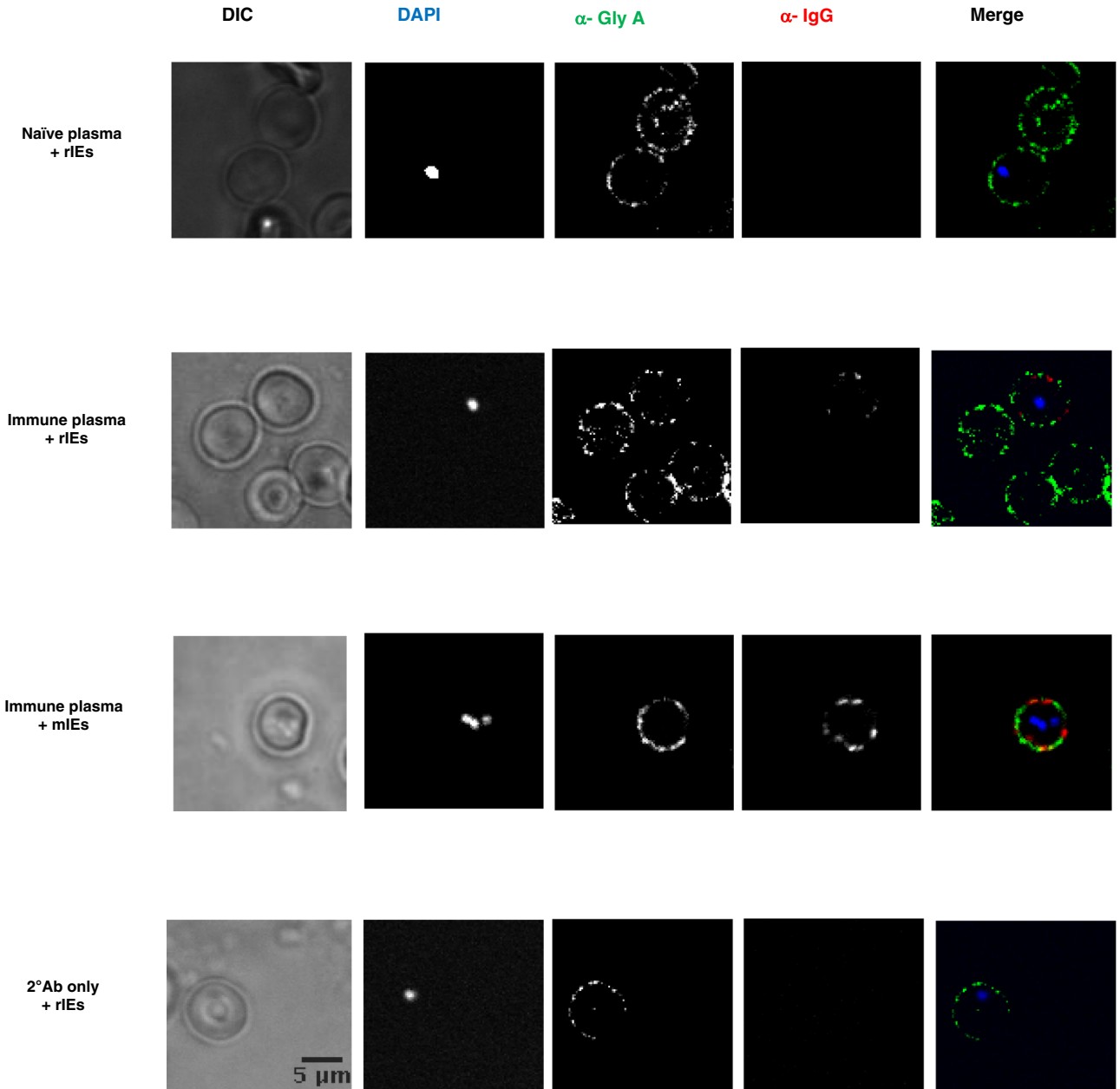

**Fig. 1 Detection of antibody binding to ring-infected erythrocytes (rIEs).** Malaria-immune plasma shows modest to low levels of binding to rIEs in comparison to mIEs in immunofluorescence assays. No binding detected with malaria-naive serum or with the secondary antibody in the absence of malaria-immune plasma. Representative data from 3 independent experiments with similar results. Source data are provided as a Source Data file.

PCR + ve groups. A subgroup analysis revealed that febrile volunteers had significantly lower levels of opsonic phagocytosis of rIEs than non-febrile, PCR + ve and PCR-ve volunteers ($P = 0.0012$, $P < 0.0001$ and $P = 0.0003$, respectively; Fig. 3B). Moreover, the time to detection of clinical malaria (study end-points) after the tenth day post-challenge was longest in volunteers with high compared to low levels of phagocytosis, $P = 0.0001$; Fig. 3C. Similarly, in a cox regression analysis, high levels of phagocytosis were strongly associated with protection, hazard ratio (HR) 0.24 (95% confidence interval 0.12–0.47), $P < 0.001$. Low levels of lumefantrine (below the minimum inhibitory concentrations[15] were detected in the C-1 samples and this potential confounding effect was accounted for in the analysis. As volunteers had been recruited from locations in Kenya that differed with respect to the malaria transmission intensity, the analysis was also adjusted for location of residence, Table 1.

**Phagocytosis of uninfected erythrocytes from ring-cultures also predicts protection against malaria.** Our results with uninfected erythrocytes closely mirrored those with ring-infected erythrocytes from ring cultures. Volunteers that were not treated had significantly higher phagocytosis of uEs compared to those that were treated ($P < 0.0001$; Fig. 3D). Febrile volunteers had significantly lower opsonic phagocytosis of uEs compared to those that did not develop fever, and those that were PCR + ve or PCR-ve ($P = 0.0054$, $P < 0.0001$ and $P < 0.0001$, respectively; Fig. 3E). Similarly, the time to detection of clinical malaria (study endpoints) after the tenth-day post-challenge was longest in volunteers with high compared to low levels of phagocytosis ($P < 0.0001$; Fig. 3F). Interestingly, the proportion of uninfected erythrocytes (uEs) from ring cultures taken up by THP-1 monocytic cells was greater than that for rIEs ($P < 0.0001$; Supplementary Fig. 3). However, this difference is lost when the

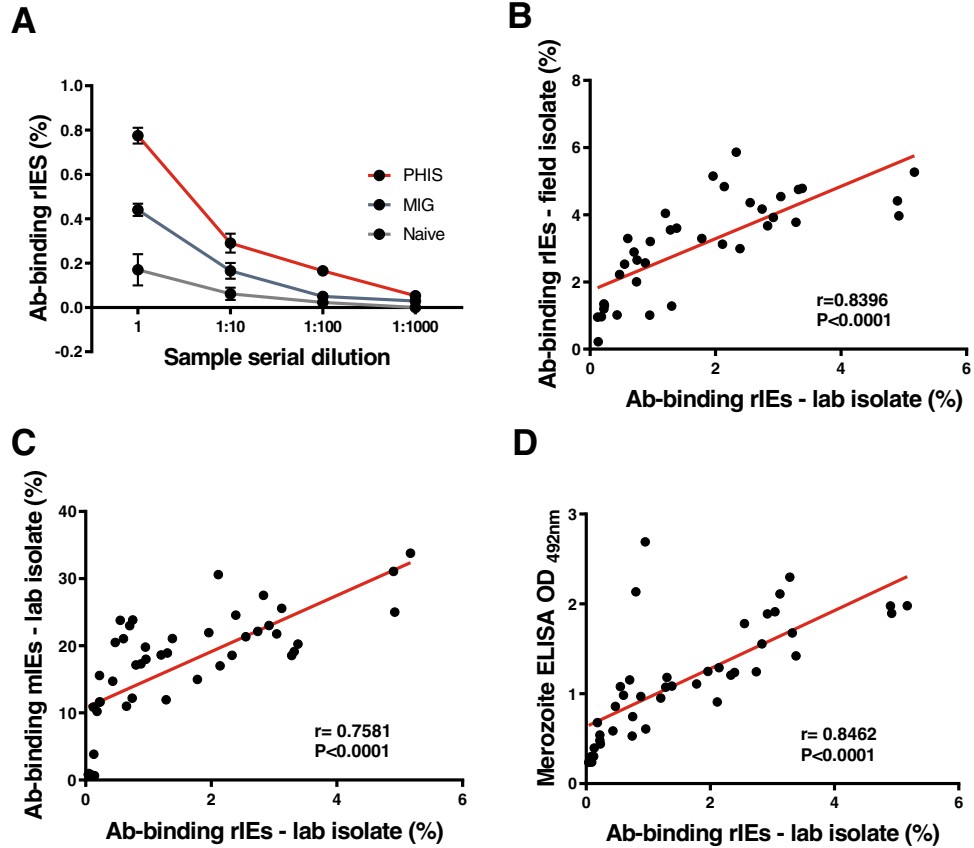

**Fig. 2 Characteristics of antibodies binding to rIEs. A** IgG antibody binding to rIEs was dose-dependent in human control samples. Error bars show the mean and standard error of the mean (SEM). $N = 3$ biologically independent controls were analyzed in duplicate in $n = 2$ independent experiments. Positive controls were pooled hyperimmune serum (PHIS, red lines) and malaria-immune globulin (MIG, blue lines), while negative controls were pooled malaria-naive serum (naive, black lines). IgG antibody binding to rIEs was strongly correlated between **B** field and laboratory isolates, as well as with IgG binding to **C** mIEs and **D** merozoites. Each black dot in 2B-D shows the level of antibody binding in biologically independent human samples ($n = 37$) from the Junju cohort to rIEs from laboratory or field isolates or to merozoites. Laboratory isolate refers to the FCR-3 strain of *P. falciparum*. Merozoites were of the NF54 strain used in the CHMI study. Correlations from single experiments conducted with $n = 2$ technical replicates per sample per assay in 2B-D were tested using Spearman's R. Source data are provided as a Source Data file.

proportion of uEs is normalised to the parasitaemia in the ring culture. A cox regression analysis revealed that the phagocytosis of both rIEs and uEs was the strongest predictor of protection than either parameter considered singly (Hazard ratio 0.18 (95% confidence interval 0.09–0.37), $P < 0.001$, Table 1). This effect remained significant after adjusting for the potential confounding effect of low lumefantrine levels and location of residence, Table 1. Of note, we found a significant positive correlation between the phagocytosis of rIEs and uEs in all samples (Spearman $r = 0.6$, $P < 0.0001$). Interestingly, this correlation was lower in untreated (protected, $r = 0.4$, $P < 0.0001$) than treated (unprotected, $r = 0.7$, $P < 0.0001$) individuals, supporting the notion that some antigens are shared between rIEs and uEs, while others are different, with the latter is more evident in the protected group.

**Merozoite proteins are the targets of antibody-dependent opsonic phagocytosis of rIEs and uEs.** We undertook a series of independent experiments to test our hypothesis that merozoite proteins were the antibody targets of the opsonic phagocytic activity we detected on rIEs and uEs. We began by testing whether peri-invasion cultures that are known to be enriched for merozoite proteins promoted the phagocytosis of uEs. We then examined the correlation of phagocytic activity against rIEs and uEs with total IgG binding to merozoites. This was followed by

phagocytosis competition assays using a selection of soluble recombinant merozoite proteins. Finally, we conducted surface mass spectrometry on rIEs and uEs and conclusively identified merozoite proteins. These results are presented sequentially in the sections below.

**Peri-invasion, but not mid-cycle culture supernatants promote opsonic phagocytosis of uEs.** Peri-invasion cultures are known to be enriched with merozoite proteins that are shed at invasion. We compared the ability of culture supernatants taken at different time points to promote the phagocytosis of uEs. We compared supernatants collected around the time of invasion versus mid-cycle (10–34 h post-invasion) and tested fresh media as a control. We found that peri-invasion but not mid-cycle cultures promoted phagocytosis (Fig. 4A). This supported our hypothesis that parasite proteins released during the invasion process trigger opsonic phagocytosis of uEs.

**Phagocytosis of rIEs and uEs is strongly correlated with merozoite IgG ELISA.** Opsonic phagocytosis of rIEs and uEs was strongly and positively correlated with total IgG against merozoites. The correlation was particularly strong for the cytophilic subclasses IgG1 and IgG3 (Spearman r between 0.6 and 0.7, $P < 0.0001$) and weaker for non-cytophilic IgG2 and IgG4

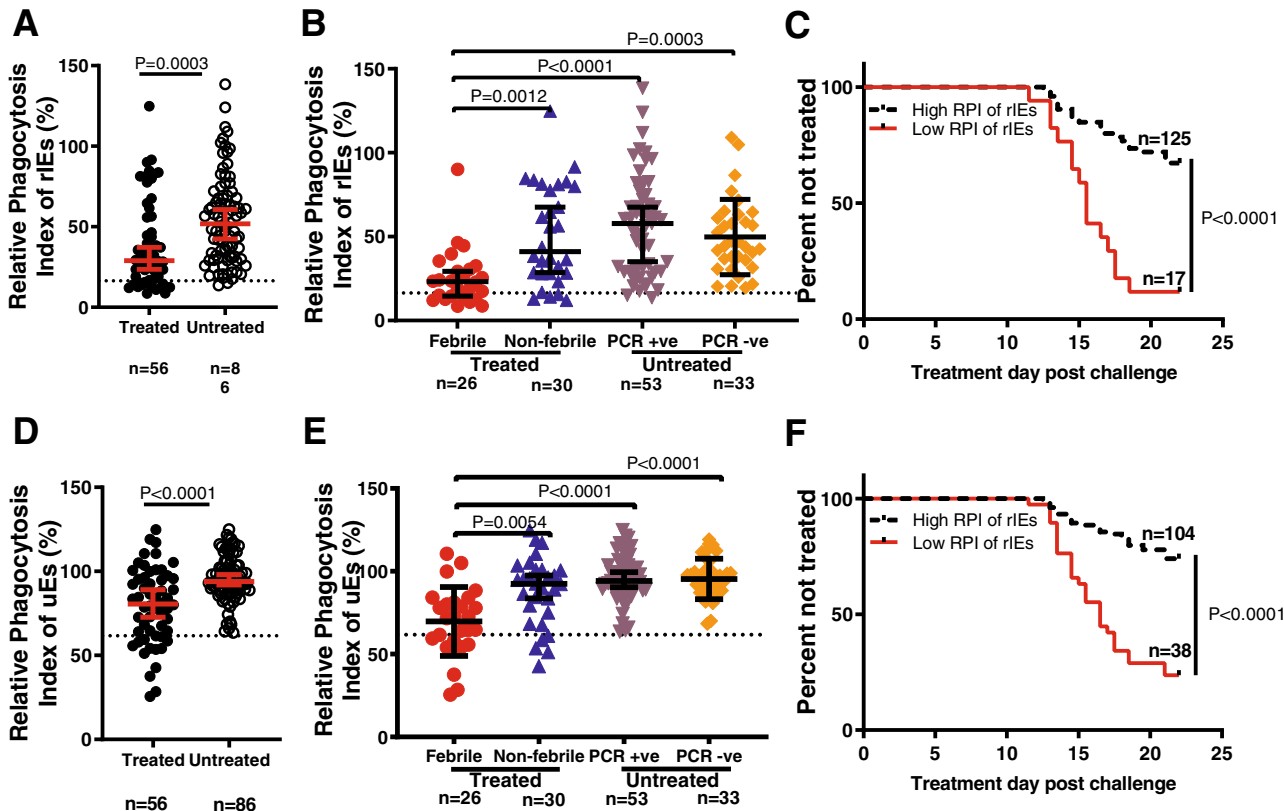

**Fig. 3 Opsonic phagocytosis of ring-infected and uninfected erythrocytes predicts protection.** The relative phagocytosis index (RPI) for ring-infected erythrocytes (rIEs) **A** or uninfected erythrocytes (uEs) **D** was compared between plasma samples from volunteers that did ($n = 56$, closed circles) or did not ($n = 86$, open circles) require treatment after a malaria challenge. Data points represent a total of $N = 142$ biologically independent samples tested in a single experiment with two technical replicates. The red lines indicate the median and 95% confidence intervals. Dotted horizontal lines indicate the seropositivity level (mean plus three standard deviations of malaria-naive plasma) for phagocytosis. $P$ value determined by two-tailed Mann–Whitney $U$ test with no adjustment for multiple comparisons. Sub-group analysis for rIEs (**B**) and uEs (**E**) respectively for treated volunteers who either developed fever (febrile, $n = 26$, red circles) or did not (non-febrile, $n = 30$, blue triangles), and for untreated volunteers in whom parasites were either detected by PCR (PCR + ve, $n = 53$, purple triangles) or remained negative (PCR-ve, $n = 33$, orange diamonds). Data points represent a total of $N = 142$ biologically independent samples tested in a single experiment with two technical replicates. The black lines indicate the median and 95% confidence intervals. Dotted horizontal lines indicate the seropositivity level for phagocytosis. $P$ values were determined by Kruskal–Wallis two-sided test followed by Dunn's multiple comparison test. **C** & **F** Kaplan–Meier curves for the time to treatment for volunteers with a high ($n = 125$, black line) versus low ($n = 17$, red line) RPI against rIEs, and a high ($n = 104$, black line) versus low ($n = 38$, red line) RPI against uEs, respectively. A total of $N = 142$ biologically independent samples were analysed in $n = 2$ independent experiments with $n = 2$ technical replicates per experiment. $P$ value determined by log-rank test in a Kaplan–Meier survival analysis and not adjusted for confounding variables. Source data are provided as a Source Data file.

**Table 1 Hazard ratios (HR) and 95% Confidence intervals (CI) from a Cox regression analysis comparing high versus low levels of phagocytosis against rIEs, uEs or both.**

| Parameter | HR (95% CI) | $P$ value | HR# (95% CI) | $P$ value | HR## (95% CI) | $P$ value |
|---|---|---|---|---|---|---|
| rIEs OPA only | 0.20 (0.11–0.37) | <0.001 | 0.23 (0.12–0.44) | <0.001 | 0.24 (0.12–0.47) | <0.001 |
| uEs OPA only | 0.22 (0.13–0.37) | <0.001 | 0.23 (0.13–0.39) | <0.001 | 0.15 (0.13–0.46) | <0.001 |
| rIEs or uEs OPAˆ | 0.53 (0.25–1.10) | 0.088 | 0.61 (0.28–1.32) | 0.166 | 0.52 (0.24–1.17) | 0.114 |
| rIEs + uEs OPAˆˆ | 0.15 (0.08–0.29) | <0.001 | 0.17 (0.09–0.34) | <0.001 | 0.18 (0.09–0.37) | <0.001 |

ˆHigh levels of phagocytosis of either rIEs or uEs was compared against low levels. ˆˆHigh levels of phagocytosis of both rIEs and uEs was compared against low levels. Hazard ratios were adjusted for #low levels of lumefantrine and ##the location of volunteer recruitment in addition. Cox regression analysis was conducted in STATA™ version 15.1. No adjustments were made for multiple comparisons. Source data are provided as a Source Data file.

(Spearman $r$ between 0.2 and 0.4, $P < 0.0001$; Fig. 4B; Supplementary Fig. 4).

**Merozoite antigens lower phagocytosis in competition assays.** We randomly selected two samples each from CHMI volunteers in three strata: those that had high (18A0012 and 16K0036), medium (17K0005 and 17K0028) and low (17K0074 and

17K0084) level IgG responses to merozoites for competition phagocytosis assays. Plasma samples were pre-incubated individually with either single or a pool of recombinant merozoite antigens (Merozoite surface proteins −1, −2 and −3 (MSP-1/−2,−3); Erythrocyte binding antigen −175 (EBA-175) and Apical membrane antigen −1 (AMA-1)) prior to opsonic phagocytosis. These antigens were selected because they are well-characterised merozoite proteins that were readily

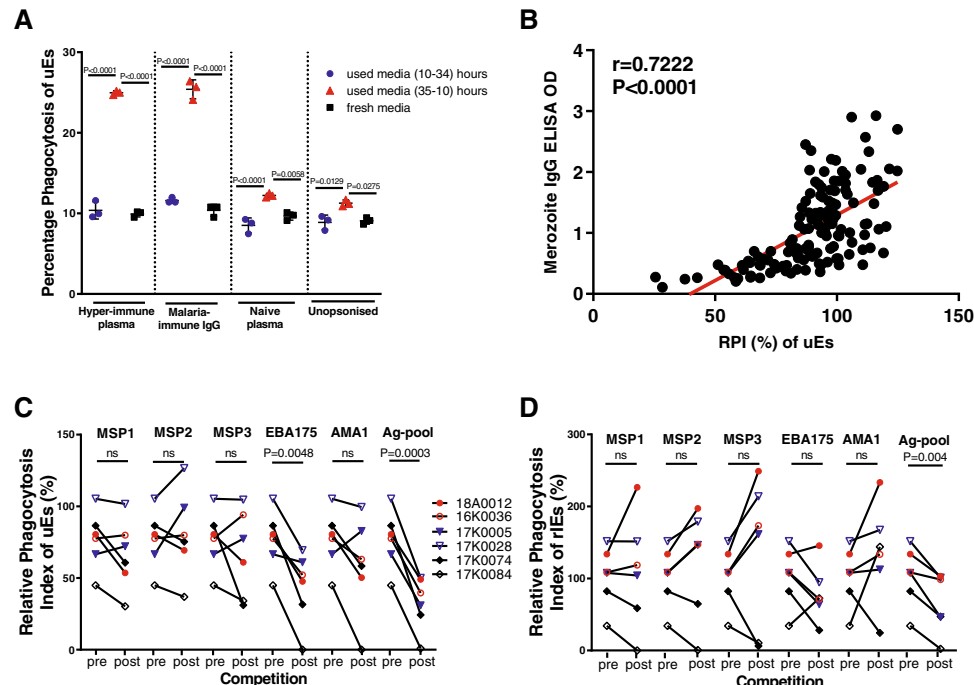

**Fig. 4 Merozoite antigens are likely targets of antibody-dependent opsonic phagocytosis of ring-infected and uninfected erythrocytes. A** Phagocytosis of freshly isolated uEs that were pre-incubated with supernatants from mid-cycle parasite cultures (10 h to 34 h, blue circles), peri-invasion cultures (35 h to 10 h post invasion, red triangles), and fresh culture media (black squares). *P* values from a one-way ANOVA test followed by Tukey's multiple comparison test. The graph shows the mean and error bars indicate the standard deviation. Each data point represents one of three biological replicates. **B** Phagocytosis of rIEs is strongly correlated with total IgG antibody binding to merozoites ($n = 142$). Each black dot indicates the level of IgG binding to either merozoites or RPI (%) of uEs in a total of $N = 142$ biologically independent samples, tested in a single experiment in duplicate. Correlation is tested using Spearmans' R. The RPI of **C** uEs and **D** rIEs using individual plasma samples ($n = 6$) before (pre) and after (post) competition with individual and pooled recombinant merozoite antigens. Coloured symbols denote individual samples with high ($n = 2$, red), medium ($n = 2$, blue) and low ($n = 2$, black) levels of IgG against merozoites, respectively. *P* values were determined using a two-tailed paired *t*-test. Each data point represents the RPI measured once for each plasma sample with two technical replicates. RPI, relative phagocytosis index. Source data are provided as a Source Data file.

available. A 19 Kda fragment of MSP-1 (MSP-$1_{19}$) remains bound to the surface of red cells following invasion[21]. Early studies demonstrated that EBA-175 is bound to the surface of infected erythrocytes[18]. The remaining antigens are not known to bind to the surface of rIEs. We compared the phagocytosis index pre- and post- incubation with these merozoite antigens. The pre-competition phagocytosis index was significantly higher than the post-competition one for both rIEs and uEs ($P = 0.0003$ and $P = 0.004$, respectively; Fig. 4C, D). This reduction in phagocytosis appears to have been primarily driven by antibodies against EBA-175 (Fig. 4C).

**Mass spectrometry identifies distinct merozoite surface and secreted proteins on the surface of ring-infected versus uninfected erythrocytes.** Finally, to conclusively identify the parasite proteins targeted by human antibodies on the surface of rIEs and uEs, we undertook mass-spectrometric analyses using a range of carefully selected experimental conditions. We surface trypsinized rIEs and uEs incubated in used and fresh media in four independent biological replicates. A total of 2882 peptides and 478 proteins were identified by liquid chromatography-tandem mass spectrometry (LC-MSMS) based on a false discovery rate (FDR) cut off of 0.01 for peptides and proteins. Most of the proteins identified were of human origin ($n = 444$), while a minority were classified as belonging to *Plasmodium* species ($n = 34$, Supplementary Fig. 5).

Eleven of thirty-four identified *Plasmodium* proteins met the criteria for statistical analysis: 1) a minimum number of 2 unique peptides were identified to derive label-free quantification (LFQ)

values, and 2) LFQ values exceeded 0 in 3 out of 4 independent replicates for at least one experimental condition. Seven of these eleven proteins were differentially identified, while four of eleven were comparably identified (Supplementary Table 1). The remaining 23 proteins did not meet these stringent criteria and were either predicted to contribute to cellular and metabolic processes (16/23), host cell invasion (2/23) or remain uncharacterized (5/23, Supplementary Table 1).

The seven differentially identified *Plasmodium* proteins showed a significantly higher abundance in trypsin-shaved ring culture and trypsin-shaved uEs in used media as compared to mock-shaved ring culture and trypsin-shaved uEs in fresh media, respectively (Fig. 5A–C, Supplementary Table 2, Supplementary Fig 6). This suggests that they are present either in culture supernatants or on the surface of rIEs or uEs. Five of these seven proteins have previously been associated with erythrocyte invasion: RhopH1, RhopH2, EBA-140, EBA-175 and enolase[22–24]. Detection of the two EBA protein family members is consistent with their roles as parasite ligands binding glycophorins on the erythrocyte surface during invasion, and their reported presence in culture supernatants[18,24].

Two members of the high molecular weight protein (RhopH) complex (RhopH2 and RhopH3) were significantly more abundant in trypsin-shaved ring culture compared to trypsin-shaved uEs in spent media suggesting that these were predominantly enriched on the surface of rIEs as opposed to uEs (Fig. 5, Supplementary Table 2). Interestingly, another component of the RhopH complex (RhopH1/Clag3.1) and two members of the rhoptry-associated protein (RAP) complex (RAP1 and

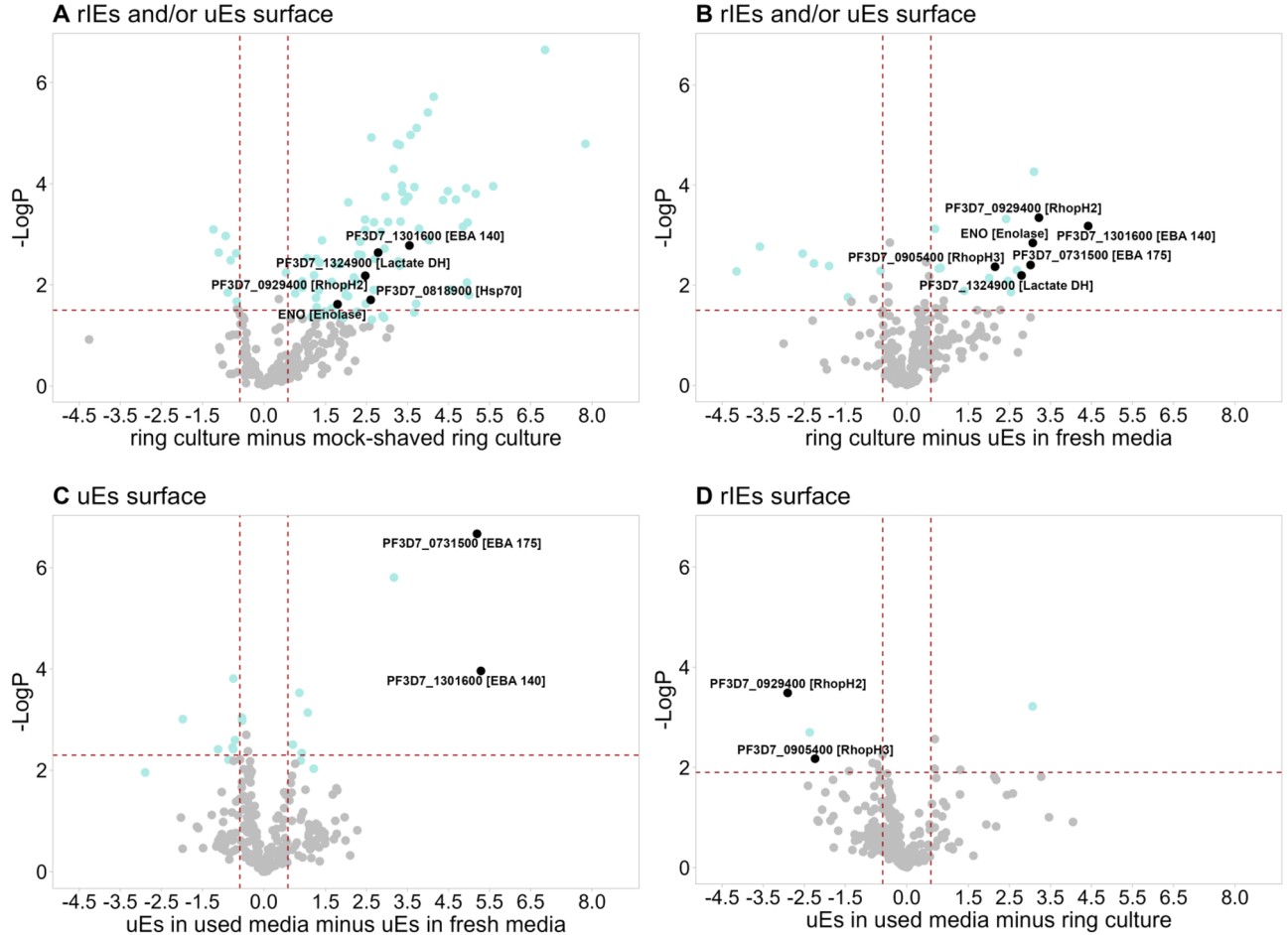

**Fig. 5 Volcano plots demonstrate significant enrichment of distinct merozoite proteins in specific experimental conditions.** Each panel shows the abundance of individual proteins after a comparative analysis of the following experimental conditions (x-axis): **A** ring cultures versus mock-shaved ring cultures, **B** ring cultures versus uEs in fresh media, **C** uEs in used media versus uEs in fresh media, and **D** uEs in used media versus ring cultures. Black and green dots in each panel indicate the *Plasmodial* and human proteins, respectively that were detected significantly more abundantly in one experimental condition compared to the other. Grey dots indicate proteins that were not detected differentially in the respective experimental conditions. For each panel, subheadings are the suggested localisation of the *Plasmodium* proteins shown in black. The negative Log10 *p*-values are displayed on the y-axis and the Log2-fold changes are displayed on the x-axis. The dotted red lines indicate the 0.05% FDR significance cut-off. Differences between pairs of treatment conditions were calculated using two-sample *t*-tests without correction for multiple comparisons. Four independent biological replicates were analysed for each experimental condition. Source data are uploaded onto the PRIDE database.

RAP2) were also identified predominantly in trypsin-shaved ring-culture as compared to other experimental conditions but did not meet the selection criteria for statistical analysis (Supplementary Tables 1 and 2). Rhoptry protein 2 (RAP2) was named ring surface protein, RSP2 when it was first discovered[4,25].

**Phagocytosis of rIE and uEs associated with significantly higher haemoglobin**. A potential and worrying consequence of the phagocytosis of rIEs and uEs is that it may predispose to anaemia[10,11]. Although haemoglobin levels remained within the normal range for all volunteers throughout the duration of the study, they were significantly lower overall on the day of discharge (DOD) compared to the day before the challenge (C-1) with a median of differences of −0.3 g/d (DOD vs C-1; Fig. 6A). We tested whether phagocytosis of rIEs or uIEs was associated with significantly lower haemoglobin levels. We found that the decrease in haemoglobin was limited to the volunteers who were treated, which includes the febrile and non-febrile categories (Fig. 6B, C). Volunteers that were treated had the lowest levels of phagocytosis.

## Discussion

We show that the antibody-mediated opsonic phagocytosis of ring-infected and uninfected erythrocytes is strongly correlated with the ability to limit the exponential multiplication of parasites following an experimental *P. falciparum* challenge in humans. We conduct a surface proteomic analysis of ring-stage cultures and identify distinct merozoite proteins on the surface of newly invaded versus uninfected erythrocytes. These data suggest that the early clearance of infected erythrocytes soon after invasion may be critically important for controlling parasitaemia and preventing the development of clinical malaria.

The importance of antibody-mediated phagocytosis in protection against malaria has previously been demonstrated using merozoites[20,26] and erythrocytes infected with mature stages of the parasite[27]. Sporozoites can also be targeted for clearance by opsonic phagocytosis[28], but this was not correlated with protection when analysed using circumsporozoite protein (CSP)-coated fluorescent beads[29]. Likewise, although low levels of phagocytosis against gametes were detected, this did not reduce infectivity to mosquitoes[30].

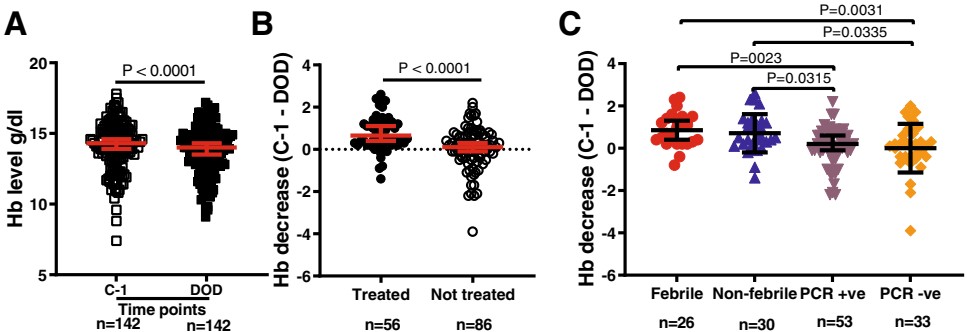

**Fig. 6 Association of phagocytosis of rIEs and uEs with high haemoglobin levels in CHMI individuals. A** Hb levels were measured once in biologically independent blood samples from volunteers ($n = 142$) at $n = 2$-time points on the day before the challenge (C-1, open squares) and the day of discharge (DOD, closed squares). Each data point represents the Hb level from independent volunteers at distinct time points. Red lines indicate the mean and 95% confidence intervals. $P$ value determined by a two-sided Wilcoxon signed-rank $t$ test for matched pairs. **B** Hb levels measured on DOD subtracted from those on C-1 for treated ($n = 56$, closed circles) versus non-treated ($n = 86$, open circles) volunteers. Each data point represents the difference in Hb levels between C-1 and DOD for each volunteer. Red lines indicate the mean and 95% confidence intervals. The dotted horizontal line indicates no difference between C-1 and DOD. The $P$ value comparing the decrease in Hb was determined by a two-tailed Mann–Whitney $U$ test. **C** Sub-group analysis for a decrease in Hb for treated volunteers who either developed fever (febrile, $n = 26$, red circles) or did not (non-febrile, $n = 30$, blue triangles), and untreated volunteers in whom parasites were either detected by PCR (PCR + ve, $n = 53$, purple triangles) or remained negative (PCR-ve, $n = 33$, orange diamonds). Each data point represents the difference in Hb levels between C-1 and DOD. Black lines indicate the mean and 95% confidence intervals. $P$ values comparing the decrease in Hb were determined by the Kruskal–Wallis test followed by Dunn's multiple comparison test. Source data are provided as a Source Data file.

In contrast to mIEs, rIEs do not have an abundance of parasite proteins expressed on their surface. Endogenous expression of variant antigens such as *P. falciparum* erythrocyte membrane protein 1 (*Pf*emp1) occurs on the surface of infected erythrocytes from approximately 16 h post-invasion[31]. We show that the phagocytosis of rIEs and uEs is strongly correlated with total and cytophilic IgG against merozoites, can be out-competed using recombinant merozoite antigens, and induced in fresh erythrocytes using the supernatants of peri-invasion parasite cultures. These results are consistent with previous studies that identified merozoite antigens as the targets of antibodies on rIE and uEs[3,32].

Interestingly, we observed some enhancement of phagocytosis in competition assays with single but not combinations of merozoite antigens. The most likely explanation for this is the variation in antibody levels between samples. The addition of soluble antigens results in the formation of multimeric IgG immune complexes that result in FcγR cross-linking, activation and enhanced phagocytosis[33]. In samples with high levels of antibodies, the addition of a single soluble antigen does not saturate the system, allowing additional antibodies of similar and additional specificities to mediate merozoite phagocytosis. In contrast, when multiple soluble antigens are added, although phagocytosis may be enhanced overall as a result of increased FcγR cross-linking, the system saturates leaving fewer antibodies available for phagocytosis. In samples with medium to low levels antibodies, the overall pattern is a reduction in phagocytosis with the addition of single or multiple soluble antigens, consistent with the reduction in available antibody for phagocytosis.

In a surface proteomics analysis of ring-stage cultures, we identified several known or potential merozoite targets of antibody responses. We used carefully selected controls which gave us a good indication that the proteins were either localised on newly invaded erythrocytes, also present on uEs in the ring culture or released into the culture supernatant. We detected EBA-175 and 140 in the trypsin-shaved fraction of rIEs and uEs in spent media, consistent with reports that these proteins are secreted into culture supernatants[18,24]. In keeping with early studies that localised the high molecular rhoptry proteins RhopH2 and RhopH3 to newly invaded erythrocytes[34,35], they were significantly more

abundant in trypsin-shaved ring cultures compared to trypsin-shaved uEs incubated in spent media. These data are compatible with recent functional studies indicating that the RhopH complex has a dual role in linking erythrocyte invasion to the formation of the new permeability pathway (NPP) that serves as a nutrient channel for parasite growth and proliferation[23,36,37]. Generally, these findings suggest that at least some of the merozoite proteins that localise to the surface of rIEs are distinct from those on the surface of uEs.

To-date, blood-stage antigens have failed to progress beyond Phase II clinical trials with go-no-go criteria often pegged to the detection activity in growth inhibition assays[38,39]. Antibodies against EBA-175 are a classic example[40–42]. Our data provide an additional mechanism that, at least for some antigens, may be an important correlate of protection. It is also tempting to speculate that vaccines inducing antibodies against combinations of selected antigens will be more effective at parasite clearance than their univalent counterparts. We and others have reported on the higher potential protective efficacy of combinations of antibodies[20,43,44]. Using stringent statistical criteria, we are cautiously confident that at least 5 of 34 proteins we identified localise to the surface of infected and uninfected erythrocytes. While we were not able to test all these specific proteins, depletion of merozoite proteins from sera significantly reduced opsonic phagocytosis of ring-stage cultures, and the contribution of EBA-175 was the most significant. Additional functional experiments with combinations of targeted antigens may be informative.

We detected *P. falciparum* enolase on the surface of trypsin-shaved rIEs but not trypsin-shaved uEs incubated in spent media, in keeping with a role in erythrocyte invasion[22]. The H12E1 monoclonal antibody against the [104]EWGWS[108] epitope of *P. falciparum* Enolase showed growth-inhibitory activity against two strains of *P. falciparum* and inhibited parasite growth in vivo in mice infected with either *P. berghei* or *P. yoelli*[45]. Previous studies detected antibodies against *P. falciparum* enolase in individuals naturally exposed to malaria[46]. Taken together, these data suggest that it may contribute to protection in humans and warrants further study.

Anaemia is a serious complication of malaria and is often multi-factorial[47]. While it has been suggested that the

antibody-dependent clearance of uEs may contribute significantly to anaemia, this has not been demonstrated conclusively[10,11]. We and others show that the proportion of uEs phagocytosed in ring cultures greatly exceeds that of rIEs[7,14,48]. Although this difference was lost when the proportion of uEs was normalised to the parasitaemia in the ring culture, the combination of phagocytosis against both rIEs and uEs was the strongest predictor of protection. In our study, volunteers with the highest level of phagocytosis against rIEs and uEs had the lowest decrease in haemoglobin. This suggests that the benefits of rapidly clearing rIE and uEs by phagocytosis outweighs the potential anaemia that could occur if parasite multiplication was unchecked. However, under the controlled conditions of our human challenge study, infections were curtailed when parasitaemia exceeded 500 parasites/µl. Thus, higher density parasitaemias could still tip this balance in favour of anaemia.

Our study focused solely on the contribution of the antibody-dependent phagocytosis of ring-infected and uninfected erythrocytes to protective immunity. It is likely that antibodies targeting other parasite stages, including free merozoites, also play an important role through phagocytosis or indeed other immune mechanisms. Accordingly, additional studies are being undertaken using these samples from the CHMI study to further delineate the components of protective immunity.

In conclusion, we build on the evidence that antibody-dependent opsonic phagocytosis targeting merozoite proteins on newly invaded and uninfected erythrocytes is a strong predictor of acquired immunity against *P. falciparum* malaria. The rapid clearance of parasites immediately following red cell invasion may play an important role in controlling parasitaemia and contribute to protective immunity. Vaccines incorporating multiple parasite antigens targeting this stage of the life cycle may prevent the progression to clinical malaria.

## Methods

**Study population**. Junju adults: sero-epidemiological studies on *P. falciparum* have been conducted among residents of the Junju sublocation of Kilifi county in coastal Kenya since 2005[49]. Peak malaria transmission occurs during the rainy months of May to July and November to December. Malaria transmission intensity is modest with an entomological inoculation rate (EIR) of 21.7 infective bites per person per year[50]. Samples from asymptomatic adult volunteers ($n = 37$) were utilised for assay optimisation and as positive controls for *P. falciparum* immune responses. Pooled hyperimmune serum (PHIS) from Junju adults and malaria-immune globulin (MIG[51]) were used as positive controls for assay optimisation and sample testing.

CHMI study: the recruitment of volunteers, conduct and outcome of the Controlled Human Malaria Infection in Semi-immune Kenyan Adults (CHMI-SIKA) study are published[15,52]. Briefly, Kenyan adults with varying levels of previous exposure to malaria were recruited for the study. Higher levels of malaria transmission intensity have been linked to more rapid and durable acquisition of immunity compared to those with lower intensity[53]. We excluded volunteers with sickle cell trait, as the major genetic confounder known to be associated with protection against malaria[54]. A total of 161 volunteers were inoculated with 3,200 sporozoites via direct venous inoculation and subsequently monitored daily for clinical symptoms and/or parasitaemia. The prespecified outcomes in the study protocol were the development of significant clinical symptoms with any evidence of malaria parasites by blood film positivity or parasitaemia >/= 500 parasites/µl. Treatment was provided immediately when these criteria were reached, or at 21 days post-inoculation when the study was completed. Treated volunteers were further classified into those who developed fever and those who did not (febrile versus non-febrile). Untreated volunteers were sub-classified based on parasite detection by PCR into PCR-ve and PCR + ve groups. Data from $n = 19$ volunteers was excluded from further analysis because they were subsequently found to have either antimalarial drugs in plasma ($n = 12$) or parasite genotypes other than that found in the NF54 challenge strain. The final study group included 44 females and 98 males, the average age was 28.7 years (range 18−45 years, standard deviation 7.1 years). Volunteers were compensated for participation.

**P. falciparum culture and synchronization**. Antibody-binding to the surface of rIEs was investigated using the FCR-3 laboratory strain of *P. falciparum*[3] and contrasted against that of a recently adapted field isolate from Kilifi, Kenya (P0000072, Biobank, KEMRI-Wellcome Trust Research Programme). The *P.*

*falciparum* NF54 strain was used for the opsonic phagocytosis assay (OPA) as this was identical to what had been utilised in the CHMI study[15]. The 3D7 strain of *P. falciparum* was used for the mass spectrometry experiments to facilitate peptide identification in human and *Plasmodium* genome databases. Parasites were maintained in in vitro culture through infection of human O+ erythrocytes that had been stored for less than 2 weeks of storage at 4 °C[55]. Parasitaemia was maintained between 8–10% and cultures synchronised using a combination of D-sorbitol lysis and heparin treatment[56,57]. Parasite stages were visualised using thin blood smears stained with Giemsa and distinguished by gating parameters in flow cytometry.

**Merozoite isolation**. Highly synchronous NF54 *P. falciparum* trophozoites were purified by MACS on 25 LD columns (Miltenyi Biotec) and supplied with fresh media. At the late-pigmented trophozoite to early schizont stage, the cysteine-protease inhibitor E64, 10 µM, was added for 8–12 h to allow schizont maturation without rupturing. Merozoites were released from mature schizonts by passing the culture through a 1.2 µM pore size filter (Pall GmbH) that was pre-blocked with 1% casein-PBS. The filtrate was collected and centrifuged at 4,000 g for 15 min and resuspended in 5 ml culture media. The cell density was determined by staining merozoites with dihydroethidium (DHE) for 30 min in the dark and subsequently mixed with CountBrightTM Absolute Counting Beads (Invitrogen) at a 1:100, 1:50 and 1:25 dilution. The merozoite concentration of the three dilution was quantified using the formula: the difference of merozoite count between stain and unstained condition/count of beads × dilution factor × number of beads.

**Opsonic phagocytosis assay (OPA)**. This was adapted from a previously published assay directed against merozoites[20,48]. Non-adherent THP-1 human monocytes were maintained in vitro in 150 ml culture flasks with THP-1 culture media (RPMI 1640 media supplemented with 2 mM L-Glutamine, 10 mM HEPES, 10% (v/v) foetal bovine serum and 1% pen strep (10,000 units/ml penicillin and 10,000 µg/ml streptomycin). The culture was kept in a humidified 37 °C incubator (Heraeus instruments, Hanau) and perfused with 5% (v/v) C0₂ and 95% (v/v) air. The cells were counted by a haemocytometer and cell density was maintained between $1 \times 10^5$ cells/ml to $1 \times 10^6$ cells/ml. The NF54 strain of *P. falciparum* was cultured to 10% parasitaemia. Ring cultures were centrifuged at 800 g for 4 min, before dual staining with 5 µg/ml Dihydroethidium (DHE) to stain DNA in rIEs and 2 µM Cell Trace™ Violet to stain cytoplasm of both rIEs and uEs for 20 min at 37 °C. Labelled ring cultures were subsequently opsonised with heat-inactivated (incubated at 56 °C for 30 min) CHMI plasma samples (collected the day before challenge, C-1), diluted 1:12.5, for 30 min at room temperature, and washed twice with wash buffer. A pool of heat-inactivated hyperimmune plasma from malaria-immune and naive adults were included in each plate as positive and negative controls respectively. The opsonised, dual labelled ring culture ($4.0 \times 10^6$ cells) was added to wells containing 150 µl of $2.0 \times 10^4$ THP-1 cells and incubated for 4 h at 37 °C in 5% (v/v) CO₂ and 95% (v/v) air. The THP-1 monocytes (effector) to erythrocytes (target) ratio was 1:200. The cells were centrifuged at 500 g for 4 h at 4 °C to stop the phagocytosis and resuspended in 200 µl of red cell lysis buffer at 37 °C for 8 min with frequent agitation to lyse the un-phagocytosed erythrocytes and washed twice with THP-1 culture media. The cells were then fixed in 1X paraformaldehyde (PFA) at room temperature for 30 min and finally resuspended in 200 µl wash buffer for analysis by flow cytometry. Sample acquisition was set to stop after 2,000 events (phagocytosis of rIEs) per test. Data were analysed using FlowJo software, version 10.6.1. THP-1 cells with both Cell Trace™ Violet and DHE had rIEs and those with Cell Trace™ Violet alone had uEs (gating strategy provided in Supplementary Fig. 7).

Competition phagocytosis assays were conducted as above. A selection of well-studied malaria vaccine candidates[58] that were readily available were utilised; merozoite surface proteins 1–3 (MSP1–3), erythrocyte binding antigen-175 (EBA-175) and apical membrane antigen 1 (AMA1). We randomly selected two samples each from the CHMI study that had high, medium and low levels of IgG antibodies to merozoites. Plasma (all from C-1) were incubated overnight at 4 °C with 100 µg/ml of recombinant merozoite antigens and competition was confirmed using a standardized ELISA[44]. The relative phagocytosis index was compared between non-competed and competed (preincubated with recombinant merozoite proteins) plasma samples.

Incubation of uninfected erythrocytes in used media. Uninfected erythrocytes (uEs) were incubated for 24 h in supernatants obtained from synchronised parasite cultures with similar parasitaemias but showing distinct blood stages. The first supernatant was collected between 35 to 10 h and represented the invasion period, while the second came from cultures between 10 and 34 h, covering the post-invasion life cycle.

**Merozoite based Enzyme-Linked Immunosorbent Assay (ELISA)**. To measure the anti-*P. falciparum* antibody titre: 100 µl of $5.0 \times 10^6$ merozoites/ml *P. falciparum* NF54 isolate were coated per well and incubated overnight at 4 °C. After washing, the 96 well plates were blocked with casein for 2 h at 37 °C, followed by incubation of each well with 100 µl of individual plasma sample (from C-1), diluted 1:500, for 2 h at 37 °C. A pool of hyperimmune plasma from malaria-immune adults and plasma samples from malaria-naive individuals were included in each

plate as positive and negative controls respectively. The plates were washed, and each well was incubated with either 100 µl horseradish peroxidase (HRP) conjugated polyclonal rabbit anti-human IgG antibody, specific for gamma chains, HRP at 1:2500 (Agilent (Dako), California, USA) or sheep anti-human IgG isotypes (The Binding Site, GmBH, Germany; IgG1, Ref. AP006, Lot number 426854-1; IgG2, Ref. AP007, Lot number 426686-1; IgG3, Ref AP008, Lot number 426692-1 and IgG4, Ref. AP009, Lot number 426695-1), each diluted to 1:1000, for 1 h at 37 °C. After washing, each well was incubated with 100 µl of O- phenylenediamine dihydrochloride (OPD) substrate for 30 min at room temperature. 30 µl of 1 M hydrochloric acid (HCl) was added to each well to stop the reaction and produced a very stable orange end solution whose absorbance was read at 490 nm using the CYTATION│3 imaging reader (Thermo Fischer Scientific) and analysed using the Gen5 v3.02 software.

**Mass spectrometry**. Parasite harvest: *P. falciparum* 3D7 parasite cultures were synchronised with D-sorbitol every 46 h until a 20% parasitaemia was achieved. Mature-infected erythrocytes (mIEs, 40–45 h) were purified by magnetic-assisted cell sorting (MACS) and cultured with 2x volume of fresh uEs for 12 h to allow invasion. The resultant ring culture (containing rIEs of 0–12 h) was purified by a combination of MACS and D-sorbitol to remove unwanted mIEs.

Uninfected erythrocytes: in control experiments, uEs (same batch and age as uEs used for parasite ring culture) were incubated for 24 h, either in 24 h-old culture supernatants (mixed culture containing at least 10% rIEs), or in fresh culture media.

Surface trypsinization: four experimental conditions were established each in four independent biological replicates; trypsinized ring cultures (~20% rIEs, ≤12 h); mock-shaved ring cultures (~20% rIEs, ≤12 h, not-trypsinized); trypsinized uEs in used media (see above) and trypsinized uEs in fresh media (see above). The cultures were washed gently with ice cold 1X PBS and 300 µl resuspended to make a final volume of 1 ml before digestion with 1 µg/ml of porcine-modified trypsin at 37 °C for 30 min. Trypsinization was terminated by the addition of 10 mg/ml of soybean trypsin inhibitor, and erythrocytes pelleted by centrifugation (725 g for 30 s). Mock-shaved ring-cultures were treated identically except that trypsin was omitted during incubation. The resultant supernatant was centrifuged again at 16,000 g for 3 min at 4 °C, peptides were quantified using the CYTATION│3 imaging reader (Thermo Fischer scientific) and analysed using the Gen 5 3.02 software.

Mass spectrometry: proteins (10 µg per sample) were loaded on SDS-PAGE-gel, which ran only a short distance of 0.5 cm. After Coomassie staining the total sample was cut out unfractionated and used for subsequent trypsin digestion according to a slightly modified protocol[59] on a DigestPro MSi robotic system (INTAVIS Bioanalytical Instruments AG). Peptides were loaded on a cartridge trap column, packed with Acclaim PepMap300 C18, 5 µm, 300 Å wide pore (Thermo Fisher Scientific) and separated in a three-step, 120 min gradient from 3 to 40% ACN on a nanoEase MZ Peptide analytical column (300 Å, 1.7 µm, 75 µm × 200 mm, Waters) carried out on a UltiMate 3000 UHPLC system. Eluting peptides were analyzed online by a coupled Q-Exactive-HF-X mass spectrometer (Thermo Fisher Scientific) running in data depend acquisition mode where one full scan at 120 k resolution was followed by up to 25 MSMS scans at 15 k resolution of eluting peptides at an isolation window of 1.6 m/z and collision energy of 27 NCE. Unassigned and singly charged peptides were excluded from fragmentation and dynamic exclusion was set to 60 sec to prevent oversampling of same peptides.

**Statistical analysis**
*Phagocytosis*. Differences in phagocytic levels were compared using the Mann–Whitney test, Kruskal–Wallis test with Dunn's Multiple Comparison post hoc test. Pairwise comparisons were analysed by the Wilcoxon signed-rank test or the paired *t*-test. The time to treatment between groups was compared using the Kaplan–Meier method and log-rank test. The level of phagocytosis was quantified as the relative phagocytosis index (RPI, the percentage of THP-1 human monocytes that had ingested merozoites in test sample relative to positive control[20]). The threshold level (analytical cutoff) above which phagocytosis was associated with protection was derived using maximally selected rank statistics[60]. The RPI was classified as either high or low based on this threshold level. Seropositive and seronegative cutoffs for phagocytosis were derived as the mean plus three standard deviations of the RPI of rIEs using naive plasma. Hazard ratios were estimated using a Cox regression analysis that adjusted for i) detectable levels of lumefantrine in the sample collected one day prior to challenge, and ii) location of residence. Correlations were examined using the spearman rank correlation. The analysis was done using GraphPad PRISM version 9.1.2, STATA™ version 15.1 and R© version 3.6.1.

*Mass spectrometry*. Data analysis was carried out by MaxQuant version 1.6.3.3[61] using experiment-specific databases (*Plasmodium falciparum*; *Homo sapiens*; total entries: 131126) extracted from Uniprot.org under default settings. Identification FDR cut-offs were 0.01 on peptide level and 0.01 on protein level. Match between runs option was enabled to transfer peptide identifications across raw files based on accurate retention time and m/z. Quantification was done using a label-free quantification approach based on the MaxLFQ algorithm[62]. A minimum of 2 quantified peptides per protein was required for protein quantification. Data were further processed by in-house compiled R-scripts to plot and filter data and the Perseus software package version 1.6.6.0 using default settings for further imputation of missing values and statistical analysis[63]. To identify differentially abundant proteins, only protein groups were considered that had 3/4 none zero LFQ-values in at least one condition. If values for samples were missing, they were imputed by random values drawn from a down shifted (1.8 standard deviation) and narrowed (0.3 standard deviation) normal distribution of the according samples. Differences of two treatment conditions were calculated with two-sample *t*-tests utilising Permutation-based FDR (0.05) and an S0 value of 0.1.

Identities and quantities of proteins and peptides were compared across the four experimental conditions. These were classified as differentially identified if they were relatively more abundant in the trypsin-shaved ring culture and uEs in used media compared to either absent or relatively less abundant in uEs in fresh media and mock-shaved ring cultures (two-sample-*t*-test). They were classified as comparably identified if the LFQ intensities were not statistically different across the samples in the four experimental conditions (two-sample-*t*-test).

**Reporting summary**. Further information on research design is available in the Nature Research Reporting Summary linked to this article.

## Data availability
The study protocol and outcomes are published[15,52]. The processed human immune response data generated in this study have been deposited in the KWTRP Harvard Dataverse repository in a Source Data file and are available under restricted access because of the small number of volunteers and potential privacy concerns. The raw human immune response data are protected and are not available due to data privacy laws. The Source Data file will be available to researchers who submit forms specifying a research purpose to a Data Governance Committee (contact details and forms available on https://dataverse.harvard.edu/dataverse/kwtrp)). Response times are between 4–6 weeks. The mass spectrometry proteomics raw data have been deposited in the ProteomeXchange Consortium via the PRIDE [1] partner repository with the dataset identifier PXD033964. The peptides that were identified were searched in the Plasmodium falciparum and Homo sapiens databases at Uniprot.org under default settings. The processed mass spectrometry data and accompanying R code are open access and available at the KWTRP Harvard Dataverse repository.

## Code availability
Mass spectrometry data were analyzed using in-house R scripts that are open access and available at the KTWRP Harvard Dataverse repository.

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

## Acknowledgements

We are grateful to all the study volunteers who have participated in the CHMI-SIKA study. We are also very grateful to the study teams at the study sites in Kilifi and Ahero, the collaborating teams at Sanaria, the study investigators, and all the clinical and laboratory teams. The CHMI-SIKA study was supported by a Wellcome Trust grant

(107499) and sponsored by the University of Oxford. This work was supported by a Sofja Kovalevskaja Award from the Alexander von Humboldt Foundation (3.2 − 1184811 - KEN - SKP) and an EDCTP Senior Fellowship (TMA 2015 SF1001) which is part of the EDCTP2 programme supported by the European Union awarded to F.H.A.O. F.K.M. was supported by a scholarship from the German Academic Exchange Service (DAAD), Funding Programme 57214224, ST-32-PKZ 91608705. K.M was supported by an NIHR Global Health Research Unit grant number 16/136/33; Tackling Infections to Benefit Africa (TIBA). K.M was also supported through the DELTAS Africa Initiative Grant No. 107754/Z/15/Z-DELTAS Africa SSACAB and from DELTAS Africa Initiative [DEL-15-003]. The DELTAS Africa Initiative is an independent funding scheme of the African Academy of Sciences (AAS)'s Alliance for Accelerating Excellence in Science in Africa (AESA) and supported by the New Partnership for Africa's Development Planning and Coordinating Agency (NEPAD Agency) with funding from the Wellcome Trust [107769/Z/10/Z] and the UK government. The views expressed in this publication are those of the author(s) and not necessarily those of AAS, NEPAD Agency, Wellcome Trust or the UK government".

## Author contributions

F.H.A.O. and F.K.M. conceived the study and wrote the paper with contributions from I.N.N., R.F. and M.S. F.K.M., I.N.N., V.K., R.F., M.S., K.M., D.O.O., M.R., K.F., D.K., J.T., P.N., M.H. and M.C.K. performed experiments and analyzed results. H.W. and M.S. helped to design and analyze the mass spectrometry experiment and provided helpful discussions. F.K.M., I.N.N., D.O.O., R.F., J.T. and K.M. prepared the figures and tables. All authors read and approved the final manuscript.

## Competing interests

B.K.L.S., Y.A., P.F.B., S.L.H., E.R.J., TR. are salaried, full-time employees of Sanaria Inc., the manufacturer of Sanaria PfSPZ Challenge. Thus, all authors associated with Sanaria Inc. have potential conflicts of interest. All other authors declare no competing interests.

## Ethics statement

Ethical approval for the Junju study was provided by the Kenyan National and Scientific Ethics Review Committee protocol number KEMRI/SERU/CGMR-C/022/3149. The CHMI study was conducted at the KEMRI Wellcome Trust Research Programme in Kilifi, Kenya with ethical approval from the KEMRI Scientific and Ethics Review Unit (KEMRI//SERU/CGMR-C/029/3190) and the University of Oxford Tropical Research Ethics Committee (OxTREC 2–16). All participants gave written informed consent. The study was registered on ClinicalTrials.gov (NCT02739763), conducted based on good clinical practice (GCP), and under the principles of the Declaration of Helsinki.

## Additional information

## CHMI-SIKA Study Team

Abdirahman I. Abdi[2], Yonas Abebe[8], Philip Bejon[2,9], Peter F. Billingsley[8], Peter C. Bull[10], Zaydah de Laurent[2], Mainga Hamaluba[2], Stephen L. Hoffman[8], Eric R. James[8], Melissa C. Kapulu[2], Silvia Kariuki[2], Domitila Kimani[2], Rinter Kimathi[2], Sam Kinyanjui[2,11,12], Cheryl Kivisi[12], Johnstone Makale[2], Kevin Marsh[2,9], Khadija Said Mohammed[2], Moses Mosobo[2], Janet Musembi[2], Jennifer Musyoki[2], Michelle Muthui[2], Jedidah Mwacharo[2], Kennedy Mwai[2,5], Francis Ndungu[2], Joyce M. Ngoi[2], Patricia Njuguna[2], Irene N. Nkumama[1,2], Omar Ngoto[2], Dennis O. Odera[1,2], Bernhards Ogutu[11,13], Fredrick Olewe[11], Donwilliams Omuoyo[2], John Ong'echa[11], Faith H. A. Osier[1,2,7✉], Edward Otieno[2], Jimmy Shangala[2], Betty Kim Lee Sim[8], Thomas L. Richie[8], James Tuju[2], Juliana Wambua[2] & Thomas N. Williams[2,14]

[8]Sanaria Inc., Rockville, MD, USA. [9]Centre for Tropical Medicine and Global Health, Nuffield Department of Medicine, University Oxford, Oxford, UK. [10]Department of Pathology, University of Cambridge, Cambridge, UK. [11]Centre for Clinical Research, Kenya Medical Research Institute, Kisumu, Kenya. [12]Pwani University, P. O. Box 195-80108 Kilifi, Kenya. [13]Center for Research in Therapeutic Sciences, Strathmore University, Nairobi, Kenya. [14]Department of Medicine, Imperial College London, London, UK.

