## [Peer Review File · Nature Communications]

Phagocytosis of Plasmodium falciparum ring-stage parasites predicts protection against malariaReviewers' Comments:

Reviewer #1:

Remarks to the Author:

Little is known about the role of antibody mediated-phagocytosis of ring-infected erythrocytes (rIEs) in blood-stage immunity. This study addresses this question by comparing phagocytic activity in sera from individuals that had different outcomes after CHMI. The authors find that phagocytosis of rIEs and uninfected erythrocytes (uEs) strongly correlates with protection in CHMI. Furthermore they identified antigen targets on rIEs and uEs surfaces using mass spec. While these findings are interesting, important questions remain, some of which should be addressed in the current manuscript.

While there is a strong correlation between protection and phagocytosis index, this is very unlikely to be the only correlate of protection. One can therefore only speculate about the relative contribution of antibody-mediated phagocytosis of rIEs to natural immunity.

An important question that can and should be addressed is whether the antigens that were identified by mass spec, are indeed targets of opsonizing antibodies that mediate phagocytosis. To do so, the authors should validate the identified antigens in preclinical immunization studies and determine whether raised antibodies protect in in vitro and/or in vivo models. Identification of novel vaccine candidates that protect against blood stage parasitemia would certainly be of interest to the field.

Other major comments

It is difficult to understand some parts of the results section since it is rather concise. For example, it is not clear from the results section what the phagocytosis index is, while this is a key element in the paper. I appreciate that this can be found in the methods section but shortly explaining this in the results section will help the reader understanding and valuing the experimental findings. Likewise, "the threshold" (line 118) should be explained in the main text. Furthermore it is not always clear why experiments were conducted and how different parts of the results section are connected; the results section currently reads as a list of standalone experiments.

The authors find that untreated volunteers have a higher relative phagocytic index than treated volunteers, for both rIEs and uEs. They also find that these values combined provide a stronger predictor for protection than each one individually. This suggests that antigens that mediate phagocytosis are different on uEs and rIEs, which is addressed by the Mass spec experiments. Have the authors also looked whether there is a (positive) correlation between relative phagocytic index for rIEs and uEs, in particular for individuals in the untreated (=protected) group?

Then the authors conducted competition experiments with merozoite antigens to investigate whether antibodies against these antigens mediate phagocytosis. It is not clear whether these antigens are present on the surface of uEs and rIEs. The authors should clarify this in the text, including references. This information is also relevant for the mass spec analysis that is presented later in the manuscript.

When the antigens were added as single antigens no significant decrease in phagocytosis was observed, except for EBA175 in combination with uEs. However, when a pool of antigens was added, a strong decrease in phagocytosis was observed, which suggests synergy between antibodies. Strikingly some individuals show enhancement of phagocytosis when single antigens were added but a strong reduction in phagocytosis when the pool of antigens was added. E.g. donor 18A0012 shows enhancement of phagocytosis of rIEs with each of the single antigens, but a reduction of phagocytosis when a pool of the five antigens is added. This observation should be explained.

uEs, incubated with media of 35-10 hours post invasion, but not with media from 10-35 hours post invasion, undergo efficient phagocytosis in the presence of immune antibodies. This suggests that

merozoite antigens released during egress and invasion, bind to uEs and mediate phagocytosis. To confirm that this is really the case, immune IgG binding to surfaces of uEs incubated with the different types of used media should be investigated.

Mass spec experiments identified 11 Plasmodium parasite proteins present on the surface of rIEs and uEs that were incubated with spent media. Unfortunately, data validating the presence of these antigens on the surface of rIEs and uEs is lacking and should certainly be included. This could be done with (monoclonal) antibodies raised against these antigens. Furthermore, the identified antigens do not correspond to the antigens used in the competition experiments described before, with the exception of EBA175. Please explain this discrepancy.

Minor comments

- Check that abbreviations are introduced properly throughout paper. E.g. rIEs (line 11), HR (line 121), LFQ (line 176)
- Please confirm that the phagocytosis thresholds in figure 1A and 2A are correct. All untreated samples in 2A seem to be above the threshold, but some untreated samples in figure 1A are below the threshold, while distributions for both seem similar.
- In Figure 2D open and closed symbols are used. It is not clear what the difference is between these two.
- Supplementary Table 2. Please explain 1 vs 0 and 2 vs 0 in the Table legend.

Reviewer #2:

Remarks to the Author:

GENERAL COMMENTS

Osier examines opsonic phagocytosis (OP) of cultured rIE and uE as a correlate of protection from malaria treatment in a controlled human infection (CHMI) study involving subjects in Kenya, and identifies merozoite/invasion antigens as the targets of OP.

The study team finds that OP of rIE and uE in CHMI 1) was higher for those meeting study endpoint requiring treatment; 2) was higher in febrile treated versus non-febrile; 3) was associated to time to clinical malaria based on OP threshold stratification. OP correlated to merozoite antigen ELISA, was competed against by soluble merozoite antigens in the OP assay, and can be reproduced using uEs treated with supernatants from re-invading parasite cultures (but not mid-stage IE cultures). MS studies of E surface proteins identified numerous merozoite/invasion antigens with some differences between rIE and supernatant-treated uE that are consistent with earlier findings regarding the same antigens. The CHMI studies are a wonderful research platform, and the immune assays and MS studies appear well-done.

The major shortcoming in this study is the lack of information about study participants and a broader definition of their immune responses to understand whether the responses studied here have a disproportionate association to protection or simply reflect background differences. Ref 15 describes the study population as "up to 200 healthy adult volunteers from areas of differing malaria transmission in Kenya" suggesting that their backgrounds are widely disparate; the "Clinical Trial primary outcome reporting" file provided with the submission shows the profound differences in clinical outcomes for participants based on their geographic area. We are given none of that detail in this paper as it relates to the immune responses measured, nor any consideration for how these differences could confound analyses to associate immune responses with protection. One expects that many immune responses differ between these groups, and therefore focusing on a very narrow response can give a misleading impression for the role of that response in protection.

SPECIFIC COMMENTS

Lines 108-118—definitions need clarification. Trial endpoint is "clinical symptoms of malaria or

parasitemia >500"— the language implies that clinical symptoms without parasitemia meets the endpoint criteria, please clarify. Later mention is made of "clinical malaria"—is this different than the definition above of "clinical symptoms of malaria", and if so please define.

L114—state study day when sample was collected for OP assay

L118—clarify meaning of phagocytosis "threshold" at first use of the term in the text

L137—"that the combination of both rIEs and uEs" should clarify that this refers to "phagocytosis of both rIEs and uEs"

L215—"We conduct the first proteomic analysis of ring-stage cultures" should clarify "surface" as done elsewhere. Smit et al 2010 reported proteomic profiling of Pf ring stages.

Reviewer #3:

Remarks to the Author:

This paper by Musasia et al. shows that the uptake of ring infected erythrocytes and uninfected erythrocytes in a THP-1 assay after incubation with (semi)-immune sera is associated with protection in a prospective cohort study.

Perhaps the most striking finding of the paper is that the antibodies mediating uptake may target parasite invasion molecules, notably EBA-175 and EBA-140. These are either extensively shed and so bind to uninfected and infected erythrocytes alike or are left on the surface after invasion.

One drawback of all such studies is they struggle to establish causation – protected individuals may make many antibody responses all of which correlate with protection. However, a particular strength of this study – which the authors barely comment on -is that it provides mechanism for the protective activity of anti-EBA-175 beyond invasion inhibition which would seem exciting and novel to me.

Specifically

1. It is often stated that the time taken for a merozoite to travel from an infected erythrocyte to a new erythrocyte is very short, therefore antibodies that target the invasion machinery of the parasite have only a short window in which to act. However, this paper excitingly suggests that antibodies targeting these molecules can aid the uptake of infected erythrocytes meaning they may be effective over a longer window. Perhaps the authors disagree but that seems to me to be the single most important finding of the study, but it would probably need following up.

I would suggest an experiment in which it is determined whether anti-EBA175 mAbs added to culture after invasion is complete can allow the phagocytosis of the IEs and UEs as well. A short look at the literature shows that there are mouse mAbs to EBA-175, however these could probably be expressed as human IgG1 Abs if necessary.

2. A further concern is around generalisability. Parasites use multiple pathways for invasion and it would be good to know if the ability of the invasion machinery to bind to the surface of erythrocytes is unique to EBA-140 and EBA-175? Is it surprising that Rh5 is not there? A very clean experiment would be to use EBA-175 knockout parasites that have switched to another invasion pathway.

3. Following on from point 2, is it a concern in the mass spec that molecules such as RESA are not present? I am not an expert in ms but and perhaps it would be beyond the scope of the paper, but would it be useful to determine if EBA-175 and EA-140 can be identified on the surface with a less severe treatment than trypsinization? Detergent treatment would surely be sufficient here and might reduce background (e.g. HSP70) contamination.

4. The data are generally well done with large number of replicates, but I felt that data in figures 3 and 4 should be done with experimental replicates as well as technical replicates if not larger numbers

of samples.

5. As a general comment, the paper is quite succinct. Even with additional experiments it seems to me to be more of a brief communication than a major article.

Point-by-point response to the reviewers' comments

We are grateful to the reviewers and editors for their detailed comments that have served to substantially improve our manuscript. We have addressed the most important concerns highlighted by the editors and many of the remaining questions. The information on the immune responses in the participants raised by Reviewer 2 is now published and appropriately discussed and referenced in the manuscript. We have also dealt with the main concerns re confounding by geographical location that Reviewer 2 raised. We have dealt with all the queries raised by Reviewer 1. Please find the point by point responses below. Line numbers indicated in response to reviewer comments represent line numbers in the revised version of the manuscript with "colour highlighting" visible.

REVIEWER COMMENTS

Reviewer #1 (Remarks to the Author):

Little is known about the role of antibody mediated-phagocytosis of ring-infected erythrocytes (rIEs) in blood-stage immunity. This study addresses this question by comparing phagocytic activity in sera from individuals that had different outcomes after CHMI. The authors find that phagocytosis of rIEs and uninfected erythrocytes (uEs) strongly correlates with protection in CHMI. Furthermore they identified antigen targets on rIEs and uEs surfaces using mass spec. While these findings are interesting, important questions remain, some of which should be addressed in the current manuscript.

While there is a strong correlation between protection and phagocytosis index, this is very unlikely to be the only correlate of protection. One can therefore only speculate about the relative contribution of antibody-mediated phagocytosis of rIEs to natural immunity.

We thank the reviewer for this observation and are in complete agreement that although we identified a strong correlation between the phagocytosis index and protection, it is unlikely that this is the only correlate of protection. We have indeed undertaken extensive experiments on more established assays like phagocytosis of merozoites, antibody-dependent neutrophil burst, the fixation of C1q of complement and growth inhibition. Our focus here is a novel assay for a mechanism that has hitherto not received much attention. To address this concern, we have modified the text in the abstract (lines 16-17) and in our conclusions in the main text (lines 384-385) to be clear that we propose a contribution and not complete explanation to immunity.

An important question that can and should be addressed is whether the antigens that were identified by mass spec, are indeed targets of opsonizing antibodies that mediate phagocytosis. To do so, the authors should validate the identified antigens in preclinical immunization studies and determine whether raised antibodies protect in in vitro and/or in vivo models. Identification of novel vaccine candidates that protect against blood stage parasitemia would certainly be of interest to the field.

We thank the reviewer for this comment and concur with the editors that preclinical immunization studies are beyond the scope of the present study and would require new funding which we are exploring.

Other major comments

It is difficult to understand some parts of the results section since it is rather concise. For example, it is not clear from the results section what the phagocytosis index is, while this is a key element in the paper. I appreciate that this can be found in the methods section but shortly explaining this in the results section will help the reader understanding and valuing the experimental findings. Likewise, “the threshold” (line 118) should be explained in the main text. Furthermore it is not always clear why experiments were conducted and how different parts of the results section are connected; the results section currently reads as a list of standalone experiments.

We regret the lack of clarity and the difficulty in following our results' sections as presented in the manuscript. We were perhaps over-enthusiastic in our attempts to be concise. We have completely reorganized our results section with the following major changes. i) We have included two new sections at the beginning of the results section showing immunofluorescence assay (IFA) and flow cytometry data for IgG antibody binding to the surface of rIEs (Figure 1) and the characteristics of this binding (Figure 2). These new data are presented in the text, lines 82 -126. ii) We merged the figures showing the impact of phagocytosis of ring-infected erythrocytes (rIEs) and uninfected erythrocytes (uEs) on the clinical outcome post-challenge (Figure 3). iii) We consolidated the data supporting merozoite antigens as the targets of phagocytosis on rIEs and uEs into one new figure (Figure 4). iv) In response to the major concern of Reviewer 2, we moved what was formerly Supplementary table 1 to the main text. We expanded this table and show that the potential confounding effect of both low levels of lumefantrine and the location of residence of the volunteers was minimal.

Additional minor changes include: i) additional clarification at the beginning of each results section to place the data in context, provide more information and to help “join up” the different results sections (eg lines 130-131, 181-189, 193-194, 212-215, 232-234, 271-272) ii) provision of explanations for terms such as “phagocytosis index” at the time of first use in the opening results section (lines 134-136), and iii) elimination of the term “threshold” with regards to phagocytosis with presentation of these data as high versus low levels of phagocytosis for further clarity (lines 142-148) and in the description of the statistical methods (lines 549-558).

The authors find that untreated volunteers have a higher relative phagocytic index than treated volunteers, for both rIEs and uEs. They also find that these values combined provide a stronger predictor for protection than each one individually. This suggests that antigens that mediate phagocytosis are different on uEs and rIEs, which is addressed by the Mass spec experiments. Have the authors also looked whether there is a (positive) correlation between relative phagocytic index for rIEs and uEs, in particular for individuals in the untreated (=protected) group?

We thank the reviewer for this comment. Yes, we have examined the correlation between the relative phagocytosis index for rIEs and uEs. As the reviewer suggests, the correlation was indeed positive when all volunteers were considered ($R = 0.64$, $P < 0.0001$). Interestingly, although it was also positive for the untreated (protected) group ($R = 0.40$, $P < 0.0001$), this correlation was stronger for the treated (unprotected) group ($R = 0.7$, $P < 0.0001$). This further supports the idea that the antigens are different on rIEs and uEs in the protected individuals, whilst the high correlation in the unprotected is driven by their lack of antibodies to antigens on both targets. We have added this data into the relevant section of the manuscript (lines 174 – 178).

Then the authors conducted competition experiments with merozoite antigens to investigate whether antibodies against these antigens mediate phagocytosis. It is not clear whether these antigens are present on the surface of uEs and rIEs. The authors should clarify this in the text, including references. This information is also relevant for the mass spec analysis that is presented later in the manuscript.

We thank the reviewer for this important observation. Of the five antigens that we tested in competition assays, only the 19 KDa fragment of Merozoite surface protein-1 (MSP-1₁₉) and the Erythrocyte binding antigen (EBA-175) have been shown to be present on the surface of rIEs. This has been clarified in the text and references have been added (lines 217-225).

When the antigens were added as single antigens no significant decrease in phagocytosis was observed, except for EBA175 in combination with uEs. However, when a pool of antigens was added, a strong decrease in phagocytosis was observed, which suggests synergy between antibodies. Strikingly some individuals show enhancement of phagocytosis when single antigens were added but a strong reduction in phagocytosis when the pool of antigens was added. E.g. donor 18A0012 shows enhancement of phagocytosis of rIEs with each of the single antigens, but a reduction of phagocytosis when a pool of the five antigens is added. This observation should be explained.

We thank the reviewer for this important observation. The simplest explanation is the variation in antibody concentrations between samples which we have now described in the manuscript. Although we randomly selected two samples for each stratum of high, medium and low titres of IgG against merozoites. Donor 18A0012 that the reviewer highlights as an example in the phagocytosis of rIEs had the highest antibody titre (ELISA OD 2.925), followed by 16K0036 (ELISA OD 2.081). The readout of our phagocytosis assay is the proportion of monocytes ingesting merozoites. In our experimental setup, FcYR engagement could only be activated by multimeric IgG immune complexes (mICs) arising from opsonized soluble antigens or merozoites. This activates the downstream signaling events that lead to phagocytosis. However, our assay readout only detects phagocytosis of merozoites. In the case of enhanced phagocytosis following the addition of a single antigen, we believe that for individuals with high levels of antibodies (Figure 4D, red symbols), i) there were still abundant antibodies (including those of other specificities) available for phagocytosis of merozoites, and ii) phagocytosis was enhanced overall because of the increased availability of mICs. When multiple soluble antigens are added, although this would still increase mICs and promote FCYR cross-linking, much less antibody is available for phagocytosis of merozoites. Where antibody concentrations are medium (Figure 4D, blue symbols) or low (Figure 4D, black symbols) – the overall pattern is a reduction in phagocytosis that is primarily driven by lower levels of antibodies. We have added details on the selection of samples (lines 213 – 216), added it to the legend of Figure 4D (lines 896 – 897) and provided this explanation in the discussion (lines 310-323).

uEs, incubated with media of 35-10 hours post invasion, but not with media from 10-35 hours post invasion, undergo efficient phagocytosis in the presence of immune antibodies. This suggests that merozoite antigens released during egress and invasion, bind to uEs and mediate phagocytosis. To confirm that this is really the case, immune IgG binding to surfaces of uEs incubated with the different types of used media should be investigated.

We agree with the reviewer about this line of investigation and would highlight that these experiments have been conducted. As shown in Figure 4A, immune IgG is incubated with uEs following coated with used media from culture supernatants at the two time points. The phagocytosis that we subsequently observe can only be explained by the binding of immune IgG as the same experiment conducted with malaria naïve IgG controls in similar conditions does not have the same effect. There is certainly a marked difference in the two results. To further elaborate this, we have included immunofluorescence assay (IFA) data to the beginning of the results section. We show binding to the surface of rIEs both by IFA and flow cytometry (Figure 1). In both methods, the level of antibody binding detected was low but impressively yielded a clear signal in subsequent phagocytosis assays. We can thus anticipate that antibody detection on the surface of uEs will be of a similar or smaller quantity; nevertheless, we detected convincing antibody-dependent phagocytosis on uEs.

Mass spec experiments identified 11 Plasmodium parasite proteins present on the surface of rIEs and uEs that were incubated with spent media. Unfortunately, data validating the presence of these antigens on the surface of rIEs and uEs is lacking and should certainly be included. This could be done with (monoclonal) antibodies raised against these antigens. Furthermore, the identified antigens do not correspond to the antigens used in the competition experiments described before, with the exception of EBA175. Please explain this discrepancy.

We thank the reviewer for these comments. The discrepancy between the antigens detected in the mass spec experiments and those used in the competition assays is simply explained by i) the sequence of experiments in relation to a PhD student project and ii) by the availability of antigens. The competition assays were prompted by the observation that peri-invasion but not mid-culture supernatants promoted phagocytosis. Given that peri-invasion supernatants are known to be enriched with merozoite proteins that are shed at invasion, we reasoned that we could relatively easily test this by conducting competition assays with merozoite proteins that were conveniently available in the laboratory. In revision the manuscript, we have now reorganized our presentation of the data to reflect this; reporting first the culture supernatant data (now Figure 4A; lines 194 - 202), correlations with merozoite IgG (Figure 4B; lines 205 - 209) and subsequently the competition assays (Figure 4C and 4D; lines 213 - 229).

The mass spec experiments were conducted last, and we considered this to be the optimal/ultimate method for validating the presence of low abundance antigens on rIEs and uEs and carefully designed complex experimental controls to ensure this was done as rigorously as possible. Prior to this, we had examined antibody binding to the surface of rings by immunofluorescence (as the reviewer suggests to test with monoclonal antibodies) and flow cytometry (now included as Figure 1). As mentioned above, we observed low levels of antibody binding by both methods and were subsequently impressed that this nevertheless to significant phagocytosis in human samples. This greatly encouraged us that the low levels of antibody binding likely had physiological relevance.

Given this low abundance of antibody binding, we reasoned and still believe that mass spectrometry is the best way to validate the presence of the antigens on rIEs and uEs. Nevertheless, in line with the reviewers comments, we did IFAs using mAbs against EBA-175, both for reasons of availability and long-standing data of its presence on the surface of newly-invaded rings (Camus & Hadley 1985, Science). As has been reported by Sim *et al* 2011, we found that mAbs R217 and R218 against EBA-175 did not recognize rings. To increase our ability to analyze more cells than those observed in IFAs, we repeated the experiment using flow cytometry. As shown in Supplementary Figure 1, despite a good experimental setup as relates to positive and negative controls, we still did not detect binding using mAbs R217 and R218. Thus, we conclude that mass spectrometry provides the best opportunity to validate these antigens.

Minor comments

- Check that abbreviations are introduced properly throughout paper. E.g. rIEs (line 11), HR (line 121), LFQ (line 176)

We have double-checked that abbreviations are introduced when first used throughout the paper. We had referenced rIEs early in the introduction (line 41), Hazard Ratio (HR, line 148) and Label-free quantification (LFQ, lines 244-245).

- Please confirm that the phagocytosis thresholds in figure 1A and 2A are correct. All untreated samples in 2A seem to be above the threshold, but some untreated samples in figure 1A are below the threshold, while distributions for both seem similar.

The reviewer is completely correct, and we are grateful that this error on our part can be rectified. The dotted lines in the figure refer to seropositivity for phagocytosis and not the thresholds used for analysis. Seropositivity simply identifies samples considered positive for phagocytosis based on the mean plus three standard deviations of naïve sera using rIEs. This is typical for malaria sero-epidemiology. The threshold however, underpins the analyses of protection and is derived statistically as highlighted in the text. The figure legends have been corrected in what is now Figure 3.

- In Figure 2D open and closed symbols are used. It is not clear what the difference is between these two.

The open and closed symbols are used to differentiate the 3 populations- open diamonds represent uEs, closed diamonds, rIEs and opened circles, uEs. This has now been added to the legend of Figure 2.

- Supplementary Table 2. Please explain 1 vs 0 and 2 vs 0 in the Table legend.

This is now Table 1 in the main text. The table and legend now include symbols that explain the analysis. In brief, high levels of phagocytosis against rIEs and uEs are compared against low levels of each individually. Subsequently, the analysis considers high levels of phagocytosis against either rIEs or uEs versus low levels of phagocytosis. Lastly, high levels of phagocytosis against both rIEs and uEs are compared against low levels of both.

Reviewer #2 (Remarks to the Author):

GENERAL COMMENTS

Osier examines opsonic phagocytosis (OP) of cultured rIE and uE as a correlate of protection from malaria treatment in a controlled human infection (CHMI) study involving subjects in Kenya, and identifies merozoite/invasion antigens as the targets of OP.

The study team finds that OP of rIE and uE in CHMI 1) was higher for those meeting study endpoint requiring treatment; 2) was higher in febrile treated versus non-febrile; 3) was associated to time to clinical malaria based on OP threshold stratification. OP correlated to merozoite antigen ELISA, was competed against by soluble merozoite antigens in the OP assay, and can be reproduced using uEs treated with supernatants from re-invading parasite cultures (but not mid-stage IE cultures). MS studies of E surface proteins identified numerous merozoite/invasion antigens with some differences between rIE and supernatant-treated uE that are consistent with earlier findings regarding the same antigens. The CHMI studies are a wonderful research platform, and the immune assays and MS studies appear well-done.

The major shortcoming in this study is the lack of information about study participants and a broader definition of their immune responses to understand whether the responses studied here have a disproportionate association to protection or simply reflect background differences. Ref 15 describes the study population as “up to 200 healthy adult volunteers from areas of differing malaria transmission in Kenya” suggesting that their backgrounds are widely disparate; the “Clinical Trial primary outcome reporting” file provided with the submission shows the profound differences in clinical outcomes for participants based on their geographic area. We are given none of that detail in this paper as it relates to the immune responses measured, nor any consideration for how these differences could confound analyses to associate immune responses with protection. One expects that many immune responses differ between these groups, and therefore focusing on a very narrow response can give a misleading impression for the role of that response in protection.

The detailed information on the study participants and their outcomes are now published (Kapulu 2021, JCI Insight) and we have added this reference to the manuscript (line 67). We agree with the reviewer that there were profound differences in clinical outcome based on the region of recruitment. We clarify however that the recruitment of volunteers from these regions was deliberate and strategically designed to capture differences in malaria transmission intensity precisely because this leads to differences in immunity. Higher exposure to malaria results in the faster and more durable acquisition of immunity and is reflected by higher levels of anti-malarial antibody. We wanted to compare CHMI outcomes in individuals with differing levels of immunity. To address the concern about potential confounding, we have included geographical location in our analytical model and present those data in Table 1. We show that the two major confounders, low levels of lumefantrine (below minimal inhibitory concentrations) and geographical location have a minimal effect on the point estimates of the hazard ratio. Notably, and as described in the published paper, we had excluded

participants with sickle cell trait as the major genetic factor known known to confer protection against malaria. We had also excluded other major infections such as HIV that may have impacted the outcomes (detailed in Kapulu 2021). We Thus, we are confident that the differences in outcomes are due to differences in immunity specifically to malaria. We have now referenced this paper, and clarified these important points in the study population section (lines 404 - 407)

SPECIFIC COMMENTS

Lines 108-118—definitions need clarification. Trial endpoint is “clinical symptoms of malaria or parasitemia >500”— the language implies that clinical symptoms without parasitemia meets the endpoint criteria, please clarify. Later mention is made of “clinical malaria”—is this different than the definition above of “clinical symptoms of malaria”, and if so please define.

This has been clarified; the trial endpoint is clinical symptoms of malaria with evidence of malaria parasites (lines 132-134). We have also clarified that clinical malaria refers to the study endpoints (line 144-145).

L114—state study day when sample was collected for OP assay

The sample was collected on the day before challenge (C-1) – clarified in line 135

L118—clarify meaning of phagocytosis “threshold” at first use of the term in the text

This has been simplified and is referred to as high levels of phagocytosis (line 145-146). See also response to reviewer 1, major comment one

L137—“that the combination of both rIEs and uEs” should clarify that this refers to “phagocytosis of both rIEs and uEs”

This has been corrected (line 170)

L215—“We conduct the first proteomic analysis of ring-stage cultures” should clarify “surface” as done elsewhere. Smit et al 2010 reported proteomic profiling of Pf ring stages.

This has been corrected (lines 288 and 325)

Reviewer #3 (Remarks to the Author):

This paper by Musasia et al. shows that the uptake of ring infected erythrocytes and uninfected erythrocytes in a THP-1 assay after incubation with (semi)-immune sera is associated with protection in a prospective cohort study.

Perhaps the most striking finding of the paper is that the antibodies mediating uptake may target parasite invasion molecules, notably EBA-175 and EBA-140. These are either extensively shed and so bind to uninfected and infected erythrocytes alike or are left on the surface after invasion.

One drawback of all such studies is they struggle to establish causation – protected individuals may make many antibody responses all of which correlate with protection. However, a particular strength of this study – which the authors barely comment on -is that it provides mechanism for the protective activity of anti-EBA-175 beyond invasion inhibition which would seem exciting and novel to me.

We thank for the reviewer for their enthusiasm and agree that these data provide an exciting and novel additional mechanism for antibodies targeting EBA-175. In addition to our comments in this regard in the discussion, we have now added this into the abstract “A surface proteomic analysis revealed the presence of merozoite proteins including erythrocyte binding antigen-175 and -140 on ring-infected and uninfected erythrocytes, providing an additional novel antibody-mediated protective mechanism beyond invasion-inhibition”, lines 11 – 14.

Specifically

1. It is often stated that the time taken for a merozoite to travel from an infected erythrocyte to a new

erythrocyte is very short, therefore antibodies that target the invasion machinery of the parasite have only a short window in which to act. However, this paper excitingly suggests that antibodies targeting these molecules can aid the uptake of infected erythrocytes meaning they may be effective over a longer window. Perhaps the authors disagree but that seems to me to be the single most important finding of the study, but it would probably need following up.

We are pleased and agree with the reviewer that our data suggests that there is a longer window for antibody-dependent parasite clearance, beyond the targeting of the invasion-machinery. We believe our data add a valuable piece to the complex puzzle of antibody-mediated parasite clearance and agree that it opens up the opportunity for more exciting research.

I would suggest an experiment in which it is determined whether anti-EBA175 mAbs added to culture after invasion is complete can allow the phagocytosis of the IEs and UEs as well. A short look at the literature shows that there are mouse mAbs to EBA-175, however these could probably be expressed as human IgG1 Abs if necessary.

We thank the reviewer for this comment but defer to the editors' advice not to undertake further mechanistic studies at this time. We believe it would be best to express mAbs to EBA-175 as human IgG1 for ease of comparison with our human data.

2. A further concern is around generalisability. Parasites use multiple pathways for invasion and it would be good to know if the ability of the invasion machinery to bind to the surface of erythrocytes is unique to EBA-140 and EBA-175? Is it surprising that Rh5 is not there? A very clean experiment would be to use EBA-175 knockout parasites that have switched to another invasion pathway.

We agree with the reviewer that parasites uses multiple pathways for invasion and reflect that actual binding to the surface of erythrocytes has been challenging to demonstrate, hence our surface proteomics approach. For this reason, we are not entirely surprised that we did not detect PfRh5, as our methodology targeted proteins that were likely to be bound to the erythrocyte surface. PfRh5 is known to interact with other proteins forming a complex that is thought to be tethered to the merozoite surface through the glycosylphosphatidylinositol (GPI) -anchored Pf113. While the idea to use EBA-175 knock out parasites is good, given the redundancy of invasion, we consider it more plausible that additional antigens beyond the EBAs contribute to the inhibition of invasion – both via receptor-ligand interactions and simply through Fc-antibody mediated mechanisms, such as the phagocytosis presented in this manuscript.

3. Following on from point 2, is it a concern in the mass spec that molecules such as RESA are not present? I am not an expert in ms but and perhaps it would be beyond the scope of the paper, but would it be useful to determine if EBA-175 and EA-140 can be identified on the surface with a less severe treatment than trypsinization? Detergent treatment would surely be sufficient here and might reduce background (e.g. HSP70) contamination.

The fact that we did not identify RESA is consistent with reports that it is present on the cytoplasmic face of human erythrocytes (Foley 1991, MBP) and thus was not detected by our trypsinization method that enriched for proteins on the external aspect of erythrocytes. We agree with the reviewer that an alternative treatment may allow us to detect this. However, this was not the purpose of our investigation which was focused on detecting antibodies binding to the surface of erythrocytes.

4. The data are generally well done with large number of replicates, but I felt that data in figures 3 and 4 should be done with experimental replicates as well as technical replicates if not larger numbers of samples.

We have double-checked our data and reorganized figures 3 and 4 into a new single figure 4. We clarify that we conducted 3 biological (experimental) replicates for 4A and not technical replicates as

previously indicated (lines 893 – 894). For what is now 4C and 4D, we conducted two biological replicates (line 900).

5. As a general comment, the paper is quite succinct. Even with additional experiments it seems to me to be more of a brief communication than a major article.

To address this comment, we have made major revisions and included additional data (including different experimental approaches) as outlined in the response to reviewer one, major comment one. As reviewer 3 notes, our data are novel and exciting, highlighting a new mechanism for protection that has hitherto been overlooked. The malaria research community has focused largely on invasion-inhibition as the main mechanism of action for antibodies targeting blood stage parasites. We challenge this view by providing an additional mechanism, focusing on ring-stage parasites that are highly understudied in comparison with merozoites or late-stage parasite-infected erythrocytes. We provide clear data from a well-controlled human challenge study showing that antibodies targeting these ring-stage parasites are strongly associated with protection. Previous attempts to study ring-stage parasites using samples from cohort studies were weak and unconvincing in comparison. We go a step further and conducted intricate mass spectrometry experiments, including a range of controls, biological and technical replicates, with stringent analytical criteria to demonstrate the presence of merozoite antigens on rings. We support these data with additional functional assays in competition assays and using culture supernatants. We believe that these range of experiments and the quality of data are high, and merit publication as an article.

Reviewers' Comments:

Reviewer #1:

None

Reviewer #2:

Remarks to the Author:

The authors have updated their manuscript in response to my concerns. Thank you.

Some remaining or follow-on points:

1. Regarding the CHMI studies that are now described in a new citation by Kapulu et al-- Kapulu et al JCI 2021 describe 169 subjects on their CHMI studies: (A) Nairobi (n = 27), (B) Kilifi North (n = 34), (C) Kilifi South (n = 93), and (D) Ahero (n = 15). The authors should clarify how the 142 CHMI subjects that they have included in their analyses correspond to the subsets of the 169 subjects described by Kapulu et al.

2. Line 174: "Of note, we found a significant positive correlation between the phagocytosis of rIEs and uEs in all samples (Spearman $r = 0.6$, $P < 0.0001$). Interestingly, this correlation was lower in untreated (protected, $r = 0.4$, $P < 0.0001$) than treated (unprotected, $r = 0.7$, $P < 0.0001$) individuals, supporting the notion that the antigens on rIEs are different from those on uEs." Can the authors clarify this inference? It's not clear to this reviewer whether all the correlations they cite here support their inference of differential antigen expression, or just one or some of them. In general, the highly significant correlations would make one surmise there may be shared antigens between rIE and uE.

3. Line 196: "We compared supernatants collected around the time of invasion (peri-invasion, 35 hours post-invasion to 10 hours into next invasion cycle) versus mid-cycle (10-34 hours post-invasion) and tested fresh media as a control." For the benefit of readers, please revise wording to state "We compared phagocytosis of uE after incubation in supernatants collected around the time of invasion..." or similar.

4. Line 310: Suggest that the sentence that begins "Interestingly, we observed 311 some enhancement of phagocytosis..." should start a new paragraph.

5. Finally, there should be some comment on the limitations of this study. A limitation of this study was that it only examined opsonophagocytosis as a correlate of protection in the CHMI studies. It will be important to compare this immune measure to others mooted as correlates of protection in the valuable CHMI sample set in future studies.

REVIEWERS' COMMENTS

Reviewer #2 (Remarks to the Author):

The authors have updated their manuscript in response to my concerns. Thank you.

Some remaining or follow-on points:

1. Regarding the CHMI studies that are now described in a new citation by Kapulu et al-- Kapulu et al JCI 2021 describe 169 subjects on their CHMI studies: (A) Nairobi (n = 27), (B) Kilifi North (n = 34), (C) Kilifi South (n = 93), and (D) Ahero (n = 15). The authors should clarify how the 142 CHMI subjects that they have included in their analyses correspond to the subsets of the 169 subjects described by Kapulu et al.

The Kapulu 2021 paper described 161 participants that completed CHMI. Nineteen of these were subsequently excluded from further analysis because they were subsequently found to either have antimalarial drugs in plasma (n = 12) or parasite genotypes other than that contained in the NF54 challenge strain (n = 7). Samples from the remaining 142 participants were analyzed in this study. This clarification has now been provided in the methods section, lines 469 - 472

2. Line 174: "Of note, we found a significant positive correlation between the phagocytosis of rIEs and uEs in all samples (Spearman $r = 0.6$, $P < 0.0001$). Interestingly, this correlation was lower in untreated (protected, $r = 0.4$, $P < 0.0001$) than treated (unprotected, $r = 0.7$, $P < 0.0001$) individuals, supporting the notion that the antigens on rIEs are different from those on uEs." Can the authors clarify this inference? It's not clear to this reviewer whether all the correlations they cite here support their inference of differential antigen expression, or just one or some of them. In general, the highly significant correlations would make one surmise there may be shared antigens between rIE and uE.

We agree with the reviewer that both correlations were significant and could be interpreted to suggest shared antigens between rIEs and uEs. We rather focused on the magnitude of the Spearman's r of 0.4 in the protected individuals versus that of 0.7 in the unprotected. Whilst both were significant, the lower correlation in the protected group (nearly 50% of the unprotected) suggested that perhaps the antigen recognition differed between both groups. As such, for both groups a proportion of antigens are likely to be shared, but this proportion appears to be lower for the protected group. This has been clarified in lines 220 -222.

3. Line 196: "We compared supernatants collected around the time of invasion (peri-invasion, 35 hours post-invasion to 10 hours into next invasion cycle) versus mid-cycle (10-34 hours post-invasion) and tested fresh media as a control." For the benefit of readers, please revise wording to state "We compared phagocytosis of uE after incubation in supernatants collected around the time of invasion..." or similar.

We have simplified this wording exactly as the reviewer has suggested – now line 240.

4. Line 310: Suggest that the sentence that begins "Interestingly, we observed 311 some enhancement of phagocytosis..." should start a new paragraph.

We have started a new paragraph as suggested, now line 356.

5. Finally, there should be some comment on the limitations of this study. A limitation of this study was that it only examined opsonophagocytosis as a correlate of protection in the CHMI studies. It will be important to compare this immune measure to others mooted as correlates of protection in the valuable CHMI sample set in future studies.

A paragraph on the limitations of this study in this regard has been added just before the conclusions – lines 425 – 430, pasted below

“Our study focused solely on the contribution of the antibody-dependent phagocytosis of ring-infected and uninfected erythrocytes to protective immunity. It is likely that antibodies targeting other parasite stages, including free merozoites, also play an important role through phagocytosis or indeed other immune mechanisms. Accordingly, additional studies are being undertaken using these samples from the CHMI study to further delineate the components of protective immunity”